# Unbalanced fertilizer use in the Eastern Gangetic Plain: The influence of Government recommendations, fertilizer type, farm size and cropping patterns

Md. Shofiqul Islam[1,2], Richard W. Bell[3], M. A. Monayem Miah[4], Mohammad Jahangir Alam[1] *

1 Department of Agribusiness and Marketing, Bangladesh Agricultural University, Mymensingh, Bangladesh, 2 Agricultural Economics Division, Bangladesh Sugarcrop Research Institute, Ishwardi, Pabna, Bangladesh, 3 Centre for Sustainable Farming Systems, Food Futures Institute, Murdoch University, Murdoch, Western Australia, Australia, 4 Agricultural Economics Division, Bangladesh Agricultural Research Institute, Gazipur, Bangladesh

* mjahangir.alam@bau.edu.bd

**Data Availability Statement:** All relevant data are within the paper and its Supporting information files.

## Abstract

Intensive cropping systems with diverse cropping patterns present a challenge for nutrient management on smallholder farms. Government-endorsed recommendations for crop fertilizer use are designed to assist farmers achieve profitable and balanced nutrient inputs, but it is unclear how closely farmers follow these recommendations. We identified farmers' current nutrient use gaps (overuse or underuse) relative to the Government-endorsed recommended nutrient doses in two cropping patterns in three representative Agro-ecological Zones of Bangladesh. A total of 330 farms were surveyed in 2019 from three farm size categories (referred to as large-, medium and small-scale) and their gaps in nutrient use were assessed relative to Government-endorsed Fertilizer Recommendation Guides (FRG) published in 2012 and in 2018: FRG-2012 and FRG-2018. The large- and medium-scale farms used 11–16%, 80–90% and 21–30%, respectively, over-doses of NPK in the cropping season under fully rice-based cropping pattern relative to FRG-2012 recommendations, while the over-dose levels were much lower for small-scale farms. Small-scale farms used much less than recommended S, Mg, Zn, B and organic manure (OM) rates relative to FRG-2012. The FRG-2018, which increased N and K recommendations but decreased the P recommendation for rice crops relative to FRG-2012, shows that all farms can decrease the dose of P (by 50%) while small-scale farms need to increase the dose of N (7%), K (16%), S (20%) and to apply Zn in the fully rice-based cropping pattern. On the other hand, the farms could greatly reduce NPK (19%, 86% and 44%, respectively) use while increasing S (14%), Mg, Zn, B and OM use relative to FRG-2018 in the pattern with the high-value potato crop. To increase crop profitability, enhance food security and save Government treasury in the Eastern Gangetic Plain enabling approaches are needed to effectively communicate the benefits of balanced nutrient use practices to farmers.

**Funding:** Australian Centre for International Agricultural Research (ACIAR) and Krishi Gobeshona Foundation (KGF), Bangladesh. ACIAR grant number: LWR/2016/136 & KGF grant number: (CN/FRPP):-ICP-II. The funders had no role in study design, data collection and analysis, decision to publish, or preparation of the manuscript.

**Competing interests:** The authors have declared that no competing interests exist.

## Introduction

The Eastern Gangetic Plain (EGP) which is home to the largest cluster of the world's rural poor has low agricultural productivity due to poor agronomic management practices [1]. Average yields of major crops like rice, wheat, maize and potato, are well below potential yields, in part due to unbalanced nutrient use practices by the smallholder farms, and are a drag on rural livelihoods and food security [2, 3]. Moreover, loss of unused nutrients via volatilization, leaching, denitrification or surface runoff lead to environmental impacts which in the long-run may feedback to further limit yield of crops [4]. Beside crop yield and environmental losses, excessive or overuse of fertilizers have a high cost to governments as the fertilizer sector is highly subsidized in the EGP [5].

Unbalanced fertilizer use (both over-dose and under-dose) is commonly reported, and is a major cause of yield gaps and food insecurity especially where agriculture is based on smallholder farms [6, 7]. For example, estimated average yield losses because of K-omission were around 0.62, 0.72 and 0.70 t ha$^{-1}$ for rice, wheat and maize, respectively in farmers' fields in India [8]. The application of balanced nutrient can increase yield of mustard by 50%, potato by 34–38% and rice by 17–23% compared with farmers' yield in Bangladesh [9]. On the other hand, overuse of nutrients relative to crop requirements, especially N and P, causes low nutrient use efficiency, and negative environmental impacts such as increased greenhouse gas (GHG) emission and groundwater contamination [10, 11]. Still, it remains unclear how closely farmers in the EGP follow recommended fertilizer rates especially under diverse rice-based cropping patterns.

The intensification of cropping and increase use of modern varieties have been supported by increased use of fertilizers over the last two decades in the EGP. Government-endorsed fertilizer recommendations aim to provide best fertilizer use guidelines for profitable yields on farms. But government-endorsed fertilizer recommendations often failed to achieve their goal [12–14]. Moreover, the complexity of using the recommended fertilizer dose is another challenge for farmers. For example, chemical fertilizers like di-ammonium phosphate (DAP—N & P), triple superphosphate (TSP—P & S), magnesium sulfate (MgSO$_4$—Mg & S) provide more than one nutrient element to the crop field [15]. Furthermore, organic materials (cow-dung, poultry litter, vermicompost, crop residue etc.) have variable nutrient composition which is often not accounted for in calculations.

Fertilizer price also plays a key role in deciding fertilizer rates in the EGP [5]. For example, farmers used lesser amounts of P and K-containing fertilizers, TSP, DAP and muriate of potash (MoP), compared to N-containing urea fertilizer before 2010 in Bangladesh [16]. After the government increased the subsidy on non-urea fertilizers, DAP and MoP fertilizer use increased by 120% and 39%, respectively, in fiscal year (FY) 2018–19 while urea and TSP fertilizer use increased by only 11% and 3%, respectively [17]. However, the gaps between farmers' current nutrient use and government endorsed recommendations are rarely examined especially under diverse rice-based cropping patterns. Some previous studies assessed fertilizer use gaps focusing on single crop rather than cropping patterns, but generally less attention was given to Mg, S, Zn, B and organic manure during gap assessment, and in most of the cases they overlook the losses to government treasury due to overuse of subsidized fertilizer [12, 18–20].

The north-west region of Bangladesh (NWRB) has a cropping intensity of 200%, but low agricultural productivity still hampers regional and national level food security. Average yield of major crops like monsoon rice is only about 2.6 t ha$^{-1}$ [21], which is much lower than attainable yield (5 ± 0.5 t ha$^{-1}$) [15]. The region is also vulnerable to climate variability impacts like drought due to high rainfall variability, extreme temperature (maximum temperature can

exceed 40˚C), extensive flooding during the rainy season, which adversely affecting crop yield [22]. The aim of the present study was to identify current nutrient use gaps on farms relative to the government-endorsed recommended nutrient doses in two rice-based cropping patterns in three representative Agro-ecological Zones (AEZs) of Bangladesh. Initially we hypothesized that farmers of NWRB use unbalanced rates of fertilizers and that the determination of farmers' current nutrient use gaps relative to government recommendations could be used to design interventions to promote agriculture productivity and minimize environment losses. We further hypothesis that farmers following recommended nutrient rates had higher crop yields than unbalanced nutrient users and that both the government and farmers could save substantial amount of treasury by adopting balanced nutrient use practices. To test the hypothesis, a survey was carried out in the NWRB during 2018–2019. We determined farmers' current nutrient use gaps for rice, potato and maize grown in two dominant rice-based cropping patterns after accounting for fertilizer nutrient inputs and other nutrient inputs such as manure and crop residue. The gaps in nutrient use were assessed relative to FRG-2012 (that was current at the time of the survey) and FRG-2018 (disseminated soon after the survey was completed). This paper also used focus group discussions (FGD) and key informant interviews (KII) to shed light on farmers' perceptions regarding nutrient use under cropping patterns and how they were linked to current nutrient use gaps.

## Materials and methods

### Study sites and cropping patterns

Thakurgoan, Mymensingh and Rajshahi districts in northern Bangladesh were selected to represent AEZ-1 (Old Himalayan Piedmont Plain), AEZ-8 (Young Brahmaputra and Jamuna Floodplain) and AEZ-11 (High Ganges River Floodplain), respectively. Three *Upazilas* (sub-district) from three districts namely Sadar *Upazila* from Mymensingh district (located at 24˚ 44' 39.192" N and 90˚ 24' 10.836" E), Sadar *Upazila* from Thakurgoan district (located at 26˚ 1' 55.5636" N and 88˚ 27' 38.5776" E) and Durgapur *Upazila* from Rajshahi district (located at 24˚ 22' 36.948" N and 88˚ 36' 11.052" E) were selected purposively [23]. Based on cropping pattern information, five agricultural blocks (ABs) from three *Upazilas* were selected using a probability proportion to size (PPS) sampling procedure [18]. The Department of Agricultural Extension (DAE), Bangladesh divided each union (sub-Upazila) into various segments called AB for providing extension services. Each agriculture block represented one cropping pattern.

For selecting cropping patterns, emphasis was given on crops and cropping intensity of the selected areas. Cropping intensity of Mymensingh, Rajshahi and Thakurgoan was over 200% and not much changed over the last three cropping years [21]. Two site-specific dominant rice-based cropping patterns were selected namely *irrigated rice (called Boro rice)-fallow-monsoon rice (called T. Aman rice)* and *potato-maize-monsoon rice* for the study. Among rice crops, irrigated rice and monsoon rice supply about 55% and 38% of total rice production of Bangladesh, respectively [24]. Almost all the regions of Bangladesh follow the *irrigated rice-fallow-monsoon rice* cropping pattern and it covers the highest net cropped area in Mymensingh, Rajshahi and Thakurgoan districts [25]. Based on highest net cropped area, three ABs namely *Sutiakhali*, *Pananagar* and *Jagannathpur* were selected from Mymensingh, Rajshahi and Thakurgoan districts, respectively under *irrigated rice-fallow-monsoon rice* cropping pattern [25].

The *potato-maize-monsoon rice* is a highly productive and profitable cropping pattern that covers large areas of Rangpur and Rajshahi division [25]. Based on the highest net cropped area, two ABs namely *Paurashava* and *Khochabari* were selected from Rajshahi and Thakurgoan districts, respectively to study the *potato-maize-monsoon rice* cropping pattern.

## Nutrients for gap identification

The FRG rates for N, P, K, S, Zn and B were used for gap assessment in the study. The FRG is a Government-endorsed recommendation guide published by Bangladesh Agricultural Research Council (BARC) every five years. It provides AEZ-based nutrient recommendations at certain yield goals considering cropping pattern, land topography, soil type, soil test result (if available) and residual effects of nutrients. The FRG recommended N, P, K, S and Zn for rice crops under diverse rice-based cropping patterns for AEZ-1, AEZ-8 and AEZ-11. For potato under *potato-maize-monsoon rice* cropping pattern, N, P, K, S, Mg, Zn and B and organic manure (OM) were considered for nutrient use gap assessment by the FRG, while for maize, N, P, K, S, Mg and Zn under *potato-maize-monsoon rice* cropping pattern were considered for AEZ-1 and AEZ-11. Other nutrients are provided by air, water, soil and plant residues.

## Sample design and survey administration

To collect primary data, a draft interview schedule was prepared for field testing. After pre-testing, collected information were gathered and analyzed. Then interview schedules were again rechecked and improved. Enumerators were trained before data collection. Data were collected from September 2018 to April 2019. The data was collected from the farmers through face-to-face interviews by trained enumerators under the supervision of the research team. To maintain data quality, the population of this study were those experienced farm-households who engaged with crop farming for last three years at least. Since the level of fertilizer use may vary with farm size [18, 26], data were collected purposively from three different farm sizes: (a) small-scale farms (0–2.49 acres) (b) medium-scale farms (2.50–7.49 acre) and (c) large-scale farms (7.50 acres and above) [27].

Before selecting sample respondents, a full list of farm-households by different farm sizes were prepared with the help of the Upazila Agriculture Officers (UAO). The stratified random sampling technique was followed to collect samples from different farm sizes. Then farmers were selected randomly from each farm size for conducting interviews. A total of 330 samples were collected taking 198 sample from *irrigated rice-fallow-monsoon rice* cropping pattern and 132 samples from *potato-maize-monsoon rice* cropping pattern (Table 1). Both FGD and KII were also conducted to triangulate the findings of the study (Table 1).

## Analytical technique

For determining the nutrient use gap, we followed three steps. First, nutrient use for each farmer was estimated by using Eq (1). Most of the chemical fertilizers and organic manures including crop residues contain more than one nutrient element (see S1 Table). Retained amounts of crop residues were triangulated during FGD and KII (see S2 and S3 Tables). Applied OMs were added as a source of nutrients to all the selected crops except potato because potato had specific recommendation for OM.

$$\sum_{i=1}^{n} N_{rikl} = \sum_{i=1}^{n} \left\{ X_{f_n} * C_{rf_n} + \left( \frac{X_{m_n}}{2} \right) * C_{rm_n} + \left( \frac{X_{j_n}}{2} \right) * C_{rj_n} \right\} \tag{1}$$

Where,

$N_{rkil}$ = Current use of r nutrient (kg ha$^{-1}$) in the k$^{th}$ plot by the i$^{th}$ farmer for l$^{th}$ crop

$X_{f_n}$ = Use of n$^{th}$ chemical fertilizer (kg ha$^{-1}$) containing r nutrient

$X_{m_n}$ = Use of n$^{th}$ organic manures (cow-dung, poultry litter etc.) (kg ha$^{-1}$)

$X_{j_n}$ = Amount of crop residue (kg ha$^{-1}$) retained from previous n$^{th}$ crop

**Table 1. Sample design and survey administration of the study.**

| Tools | Participants | No. of agricultural block (AB) covered | Sample distribution | Total sample size | |
|---|---|---|---|---|---|
| | | | | IFM* | PMM* |
| Field survey | Different categories of farmer | Total of 5 ABs | 66 farmers under each AB & each cropping pattern | 66x3 = 198 | 66x2 = 132 |
| | | IFM = covered three ABs from three districts; | | | |
| | | PMM = covered two ABs from two districts | (SF* = 42, MF* = 20, LF* = 4) | | |
| Focus Group Discussion (FGD) | Farmer & SAAO* | IFM = three FGDs conducted in three blocks | 10 participants in each FGD | 10x3 = 30 | 10x2 = 20 |
| | | PMM = two FGDs conducted in two blocks | (SF = 4, MF = 3, LF = 2 and SAAO = 1) | | |
| Key Informant Interview (KII) | UAO & fertilizer dealer | IFM = KII conducted in three ABs | Two participants in each AB | 2x3 = 6 | 2x2 = 4 |
| | | PMM = KII conducted in two ABs | (UAO = 1 & fertilizer dealer = 1) | | |

* SF = small-scale farms; MF = medium-scale farms; LF = large-scale farms; IFM = *irrigated rice-fallow-monsoon rice* cropping pattern; PMM = *potato-maize-monsoon rice* cropping pattern; SAAO-Sub-assistant Agricultural officer; UAO-Upazila Agricultural officer.

$C_{rf_n}$ = r nutrient composition (%) within $n^{th}$ chemical fertilizer

$C_{rm_n}$ = r nutrient composition (%) within $n^{th}$ organic manures

$C_{rj_n}$ = r nutrient composition (%) within $n^{th}$ crop residue

About 50% of the nutrients supplied by vermicompost, poultry litter or cow-dung (decompose) are assumed to be available for uptake by crops [15]. Similarly, about 50% of the nutrients left in crop residues are available for the following crop cultivated [28]. Hence we added total used chemical fertilizers (kg ha$^{-1}$) and half of the nutrient inputs applied in organic sources (vermicompost, poultry litter, cow-dung and remaining crop residues) to estimate the total rates of specific nutrient inputs (kg ha$^{-1}$) from both chemical and organic fertilizers, respectively (see S6 and S7 Tables).

Secondly, the FRG was followed to select AEZ-based nutrient recommendations under the two selected cropping patterns. The FRG-2012 recommendation was available for the farmers at the time of the survey. But the FRG-2012 was replaced by a new version of FRG (FRG-2018) which was available for the farmers soon after the field survey was completed and is used at present. To assess the existing scenario of nutrient use gaps and to provide relevant current recommendations, nutrient use gaps were assessed for each of these two FRGs.

Finally, we subtracted nutrient use for each farmer from the recommended nutrient doses to determine the farmers' current nutrient use gaps using Eq (2).

$$D_{ir} = N_{ir} - R_{ir} \tag{2}$$

Where, $D_{ir}$ = nutrient use gap of r nutrient (kg ha$^{-1}$) by the $i^{th}$ farmer; $N_{ir}$ = current use of r nutrient (kg ha$^{-1}$) by the $i^{th}$ farmer; $R_{ir}$ = recommended dose of r nutrient (kg ha$^{-1}$) for either FRG-2012 or FRG-2018. If, $D_{ij}$ = 0, that indicates no nutrient use gap; "+"indicates rates over the recommended dose of r nutrient and "-"indicates the under-doses of r nutrient.

We also estimated the financial loss of both the farmers and Government of Bangladesh because of using over-doses nutrients on the farms using Eqs (3) and (4). The loss was calculated only for NPK nutrients as fertilizers containing these three nutrients are highly

subsidized in Bangladesh.

$$F_{inl} = O_{inf} \times F_{nfp} \tag{3}$$

$$G_{inl} = O_{inf} \times G_{nfs} \tag{4}$$

Where, $F_{inl}$ = financial loss of the i[th] farmer for using over-dose of n[th] chemical fertilizer; $G_{inl}$ = financial loss of the Government for using over-dose of n[th] chemical fertilizer by the i[th] farmer; $O_{nf}$ = rates of over-dose use (kg ha[-1]) of the n[th] chemical fertilizer relative to FRG provided recommendation by the i[th] farmer; $F_{nfp}$ = maximum retail price (USD) of per kg of n[th] chemical fertilizer at the local market for the farmers; and $G_{nfs}$ = amount of Govt. subsidy (USD) for per kg of n[th] chemical fertilizer.

To calculate $G_{nfs}$, following Eq (5) was used-

$$G_{nfs} = G_{pnf} - D_{pnf} \tag{5}$$

Where, $G_{pnf}$ = the total government procurement cost (for imported fertilizers) per kg of n[th] fertilizer (USD) and $D_{pnf}$ = fertilizer dealers purchase price (USD) per kg of n[th] fertilizer (fixed by the government of Bangladesh).

The government of Bangladesh sells chemical fertilizers only to the government-appointed fertilizer dealers. For that reason, $G_{nfs}$ was estimated by deducting the total government procurement cost per kg of n[th] chemical fertilizer (for imported fertilizer) from the fertilizer dealers purchase price per kg of n[th] chemical fertilizer.

## Results

### Socio-demographic characteristics

The results of socio-demographic characteristics indicate that average landholding of the farms of the study areas was 1.1 ha (Table 2). The dominant land type was highland followed by medium highland and lowland. More than 80% farmers reported no knowledge about FRG and its recommended fertilizer doses. Low soil test use among farmers (less than 10%) indicates that farmers are making decisions about fertilizer use and rates without measurements of the current fertility status of soils (Table 2). Very few farmers (ranged between 8–17%) received soil nutrient management and crop farming related training. All categories of farmers took advice from fertilizer dealers and peer farmers (54%) regarding fertilizer use more than from Government-appointed extension agents (16%) especially with regard to the *potato-maize-monsoon rice* cropping pattern (Table 2). Only 27–32% farmers of the study areas had organizational membership status.

### Descriptive statistics of nutrient use gaps

Nutrient use rates by farms that were within ±5% deviation from nutrient use rates recommended in FRG-2012 and FRG-2018 were treated as following the recommended rates. Under selected cropping patterns, farms used under-doses, close to recommended doses and over-doses of nutrients relative to both FRG-2012 and FRG-2018 (Table 3). In general, approximately 5–41% of farms used recommended NPK under *irrigated rice-fallow-monsoon rice* cropping pattern relative to the FRG-2012 and FRG-2018 where it was only 5–18% under the rice-based pattern with high-value potato crop.

The majority of the farms used over-doses of NPK under the pattern with the high-value potato crop (Table 3). The percentage of farms that used recommended rates of nutrients was low especially for S and Zn under the *irrigated rice-fallow-monsoon rice* (ranged 2–8%) and

**Table 2. Socio-demographic characteristics of the farmers in diverse rice-based cropping patterns in the study areas.**

| Particulars | Cropping Patterns | | | | | | | |
|---|---|---|---|---|---|---|---|---|
| | Irrigated rice-fallow-monsoon rice | | | All farms (n = 198) | Potato-maize-monsoon rice | | | All farms (n = 132) |
| | Large-scale farm (n = 12) | Medium-scale farm (n = 60) | Small-scale farm (n = 126) | | Large-scale farm (n = 8) | Medium-scale farm (n = 40) | Small-scale farm (n = 84) | |
| Farm size (ha) | 4.0 | 1.6 | 0.6 | 1.1 | 3.5 | 1.5 | 0.7 | 1.1 |
| Farming experience (years) | 22.3 | 19.6 | 20.1 | 20.1 | 19.6 | 16.5 | 18.3 | 17.9 |
| Age (years) | 44.5 | 40.3 | 41.6 | 41.4 | 43.6 | 37.8 | 39.0 | 38.9 |
| Education (years) | 10.2 | 8.8 | 6.0 | 7.1 | 10.3 | 7.8 | 5.7 | 6.6 |
| Gender (%) | | | | | | | | |
| Male | 100 | 100 | 100 | 100 | 100 | 100 | 100 | 100 |
| Training on soil nutrient management (%) | 25.0 | 16.7 | 5.6 | 10.1 | 12.5 | 10.0 | 6.0 | 7.6 |
| Crop farming related training (%) | 58.3 | 41.7 | 32.5 | 36.9 | 37.5 | 20.0 | 14.3 | 17.4 |
| Land typology (%) | | | | | | | | |
| Highland | 66.7 | 51.7 | 27.0 | 36.9 | 62.5 | 57.5 | 51.2 | 53.8 |
| Medium highland | 33.3 | 38.3 | 48.4 | 44.4 | 37.5 | 42.5 | 32.1 | 35.6 |
| Lowland | | 10.0 | 19.0 | 15.2 | | | 16.7 | 10.6 |
| Very lowland | | | 5.6 | 3.5 | | | | |
| Knowledge regarding optimum dose and FRG (%) | 25.0 | 23.3 | 18.3 | 20.2 | 25.0 | 20.0 | 13.1 | 15.9 |
| Farmers tested their soil (%) | 16.7 | 18.3 | 6.3 | 10.6 | 12.5 | 5.0 | 4.8 | 5.3 |
| Decision regarding Fertilizer application (%) | | | | | | | | |
| Advice from SAAO | 50.0 | 28.3 | 23.1 | 26.3 | 25.0 | 15.0 | 15.5 | 15.9 |
| Advice from fertilizer dealers & peer farmers | 33.3 | 38.4 | 45.2 | 42.4 | 62.5 | 65.0 | 47.6 | 53.8 |
| Own experience | 16.7 | 33.3 | 31.7 | 31.3 | 12.5 | 20.0 | 36.9 | 30.3 |
| Distance of input market (km) | 2.9 | 3.2 | 3.1 | 3.1 | 2.3 | 2.6 | 2.6 | 2.6 |
| Member of any organization (%) | 58.3 | 36.7 | 27.8 | 32.3 | 37.5 | 35.0 | 21.4 | 26.5 |
| Received credit (%) | 58.3 | 43.3 | 31.7 | 36.9 | 62.5 | 52.5 | 44.0 | 47.7 |
| Livestock/household (no.) | | | | | | | | |
| Cattle | 2.4 | 2.7 | 2.6 | 2.6 | 2.1 | 2.2 | 2.5 | 2.4 |
| Goat and Sheep | 2.8 | 2.1 | 3.1 | 2.8 | 2.8 | 2.7 | 3.4 | 3.1 |
| Poultry | 10.3 | 12.2 | 12.8 | 12.4 | 11.3 | 13.5 | 12.4 | 12.6 |

Source: Field Survey, 2018–2019

*potato-maize-monsoon rice* (ranged 5–14%) cropping patterns relative to both the FRGs (Table 3). Even lower percentage of farms followed recommendations for Mg, B and OM (0% to 11% relative to FRG-2018) under *potato-maize-monsoon rice* cropping pattern.

## Nutrient use gaps under *irrigated rice-fallow-monsoon rice* cropping pattern

**Nutrient use gaps in irrigated rice.** In irrigated rice, the large-scale and medium-scale farms used over-doses of NPK relative to FRG-2012 recommendations (Fig 1A and 1B).

**Table 3. Percentage of respondent farms using over-dose, under-dose and recommended dose of nutrients relative to government-endorsed recommendations (both the fertilizer recommendation guide of 2012- FRG-2012 and of 2018 FRG-2018) under contrasting rice-based cropping patterns.**

| Nutrient | Cropping pattern | | | | | | | | | | | |
|---|---|---|---|---|---|---|---|---|---|---|---|---|
| | Irrigated rice-fallow-monsoon rice (n = 198) | | | | | | Potato-maize-monsoon rice (n = 132) | | | | | |
| | Over-user (%) | | Recommended dose user (%) | | Under-user (%) | | Over-user (%) | | Recommended dose user (%) | | Under-user (%) | |
| | FRG-2012 | FRG-2018 | FRG-2012 | FRG-2018 | FRG-2012 | FRG-2018 | FRG-2012 | FRG-2018 | FRG-2012 | FRG-2018 | FRG-2012 | FRG-2018 |
| N | 54.5 | 12.1 | 18.7 | 40.9 | 26.8 | 47.0 | 81.8 | 81.8 | 18.2 | 18.2 | | |
| P | 65.2 | 71.7 | 9.1 | 5.6 | 25.8 | 22.7 | 88.6 | 100 | 7.6 | | 3.8 | |
| K | 47.5 | 17.2 | 18.7 | 11.6 | 33.8 | 71.2 | 90.9 | 94.7 | 5.3 | 5.3 | 3.8 | |
| S | 1.5 | 24.2 | 2.0 | 7.1 | 96.5 | 68.7 | 3.0 | 24.2 | 4.5 | 14.4 | 92.4 | 61.4 |
| Zn | 56.1 | 42.4 | 7.6 | 6.6 | 36.4 | 57.6 | 1.5 | 3.8 | | 4.5 | 98.5 | 91.7 |
| Mg | | | | | | | 5.3 | 50.0 | 10.6 | | 84.1 | 50.0 |
| B | | | | | | | | 15.9 | 2.3 | | 97.7 | 84.1 |
| OM | | | | | | | | 21.2 | 0.8 | 10.6 | 99.2 | 68.2 |

Note: OM indicates organic manure. For calculating nutrient use gaps of each nutrient relative to the recommended dose, total use rates of each nutrient in all crops in the cropping pattern were subtracted from total recommended rates. Here, ±5% deviation of nutrient use from recommended nutrient dose in the cropping season was treated as a recommended dose user.

Under-dose of K and S was prevalent, especially by the small-scale farms. Despite low use of Zn containing fertilizer, small-scale farmers used over-dose of Zn in irrigated rice unlike large- and medium-scale farmers because of greater use of OMs. The FRG-2018 which increased N and K recommendations (by 30 & 11 kg ha$^{-1}$, respectively) for irrigated rice relative to FRG-2012 (see S7 Table) meaning that farms especially small-scale farms of Mymensingh, Rajshahi and Thakurgoan districts need to increase rates of N and K (by 27.9, 23.5 & 26.9 and 16.9, 12.7 & 12.3 kg ha$^{-1}$, respectively) use in irrigated rice (Fig 1C). The result also shows that farms especially large-scale farms need to reduce rates of P (by 11.4, 22.2 & 23.5 kg ha$^{-1}$, respectively) use relative to FRG-2018.

**Nutrient use gaps in monsoon rice.** Similar to irrigated rice, the result shows that large-scale and medium-scale farms used over-doses of NPK in monsoon rice relative to FRG-2012 (Fig 2A and 2B). The over-dose level of NPK was higher for farms of Rajshahi district followed by Thakurgoan and Mymensingh districts. But small-scale farms mostly used under-dose of N and K, but over-dose of P (Fig 2C). The result also shows that all categories of farms used under-dose S and Zn relative to FRG-2012. The FRG-2018 which highly increased the K recommendation (by 15 kg ha$^{-1}$) and decreased P and S recommendation for monsoon rice relative to FRG-2012 indicates that small-scale farms of Mymensingh, Rajshahi and Thakurgoan districts need to increase rates of K (by 21.6, 10.8 & 16.1 kg ha$^{-1}$, respectively) use while the large-scale farms need to decrease rates of P (by 5.7, 9.9 & 9.3 kg ha$^{-1}$, respectively) use in monsoon rice. The result also implies that small-scale farms need to increase the rates of S and Zn as per FRG-2018.

We also estimated NPKS use gaps by subtracting total use of specific nutrient from total recommended rates of that nutrients for the cropping year for the *irrigated rice-fallow-monsoon rice* cropping pattern (see S4 Table). We estimated year-wise gaps only for commonly used nutrients, NPKS. We found that farms used over-doses of NPK (by 5%, 48% and 5%, respectively) in the cropping season relative to FRG-2012 in the *irrigated rice-fallow-monsoon*

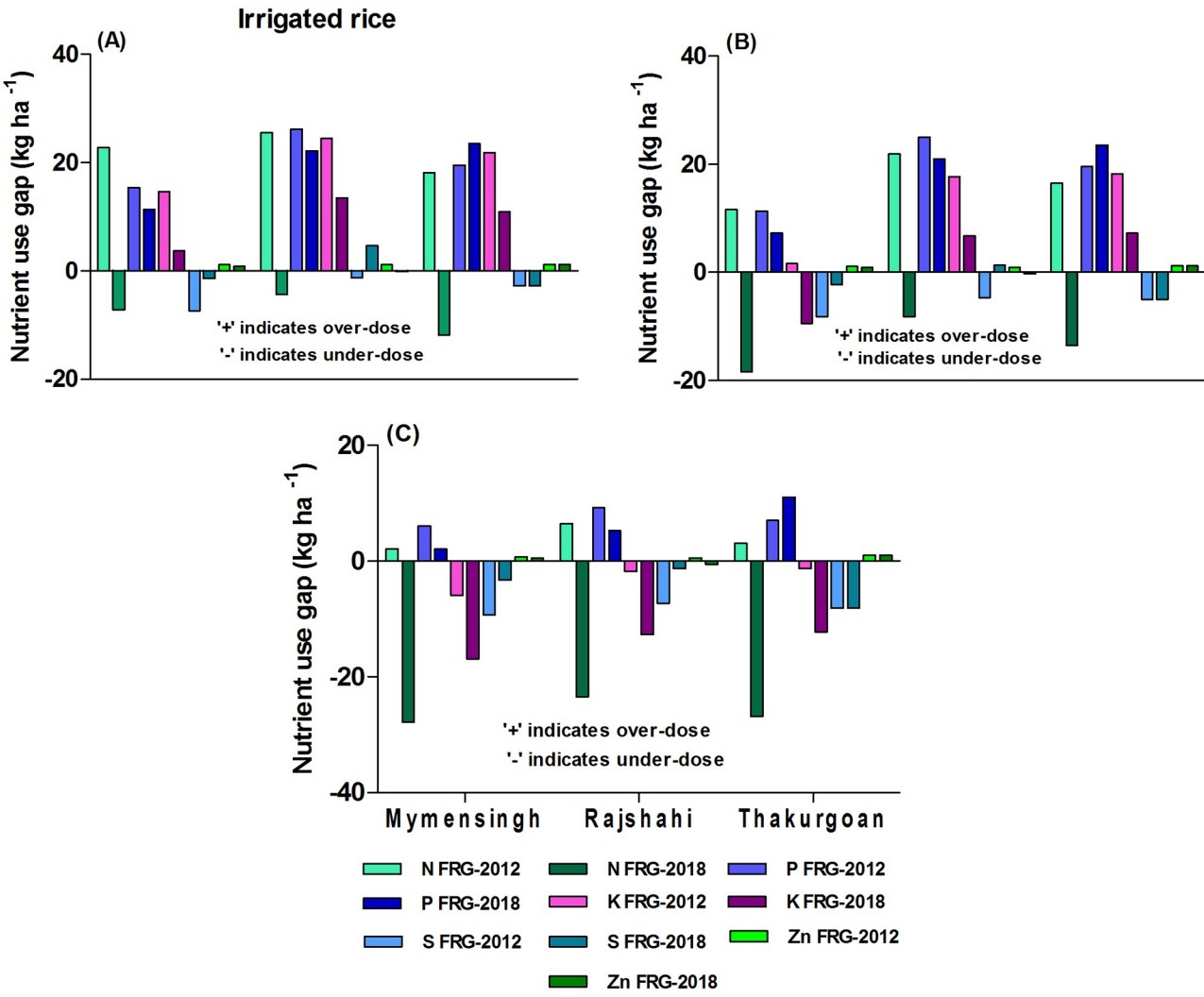

**Fig 1. Farms' current nutrient use gaps relative to FRG-2012 and FRG-2018 for irrigated rice: A) large-scale farms B) medium-scale farms C) small-scale farms for the *irrigated rice-fallow-monsoon rice* cropping pattern.**

*rice* cropping pattern but used under-dose of S (by 50%). On the other hand, farms used under-doses of NKS (by 7%, 16% and 20%, respectively) and over-dose of P (by 50%) in the cropping season relative to FRG-2018 for the *irrigated rice-fallow-monsoon rice* cropping pattern. The over-use rates of nutrients increased with increase of farm size.

### Nutrient use gaps under *potato-maize-monsoon rice* cropping pattern

**Nutrient use gaps in potato.** For potato crops, all categories of farms used high over-dose of NPKS relative to FRG-2012 (Fig 3). The over-dose level of NPKS was higher for large- and medium-scale farms than for small-scale farms. Similar to irrigated rice, farms of Rajshahi district used higher over-dose of nutrients relative to the farms of Thakurgoan district. For Mg, Zn and B, there were under-doses especially by the small-scale farms (Fig 3C). All categories of farms used high under-dose of OM in potato relative to FRG-2012 which increased with increase of farm size.

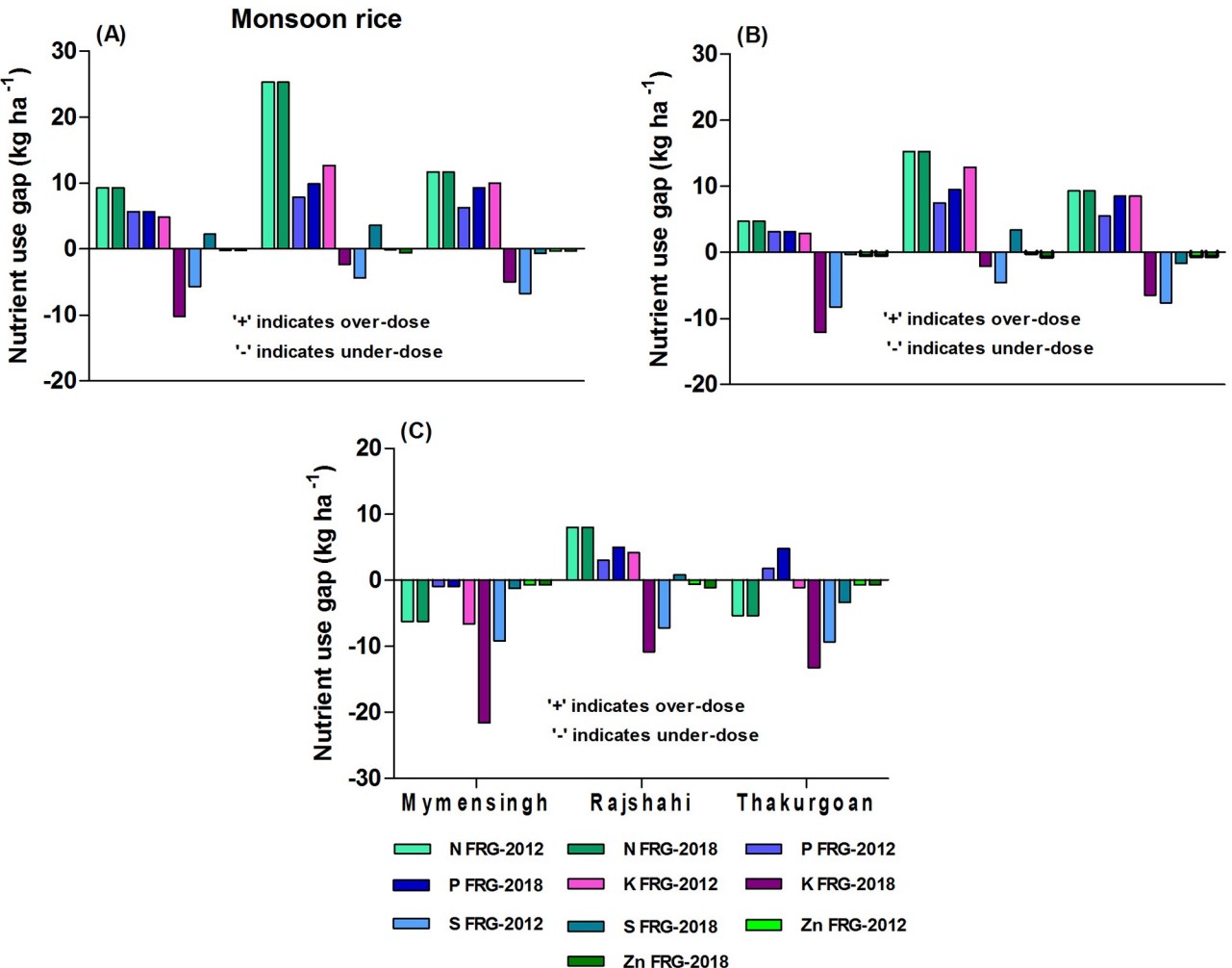

**Fig 2. Farms' current nutrient use gaps relative to FRG-2012 and FRG-2018 for monsoon rice: A) large-scale farms B) medium-scale farms C) small-scale farms for the *irrigated rice-fallow-monsoon* rice cropping pattern.**

The result as per FRG-2018 which highly reduced recommendation of P (by 10 kg ha⁻¹) and OM (by 2 t ha⁻¹) for potato relative to FRG-2012 (see S7 Table) indicates that farms especially large-scale farms of Rajshahi district can reduce rates of NPK (by 131.3, 61.3 & 92.5 kg ha⁻¹, respectively) and increase rates of OM (by 2.4 t ha⁻¹) use in potato (Fig 3A). The result as per FRG-2018 also implies that small-scale farms need to increase Mg, Zn and B use in potato.

**Nutrient use gaps in maize.** The result as per FRG-2012 shows that all categories of farms used high under-doses of nutrients in maize (Fig 4). The level of under-dose of NPK was higher for small-scale farms of Rajshahi district (by 46.1, 26.5 & 33.8 kg ha⁻¹, respectively) relative to farms of Thakurgoan district (Fig 4C). The FRG-2018 which reduced the PKS recommendations for maize relative to FRG-2012 indicates that farms especially small-scale farms of Rajshahi district need to increase rates of NPK (by 46.1, 16.5 & 19.8 kg ha⁻¹, respectively) use in maize (Fig 4C).

**Nutrient use gaps in monsoon rice.** All categories of farms used over-dose of NPK as per FRG-2012 but used under-dose of S and Zn in monsoon rice (Fig 5). The result as per FRG-2018 which increased the K recommendation (by 3.0 kg ha⁻¹) and decreased P and S

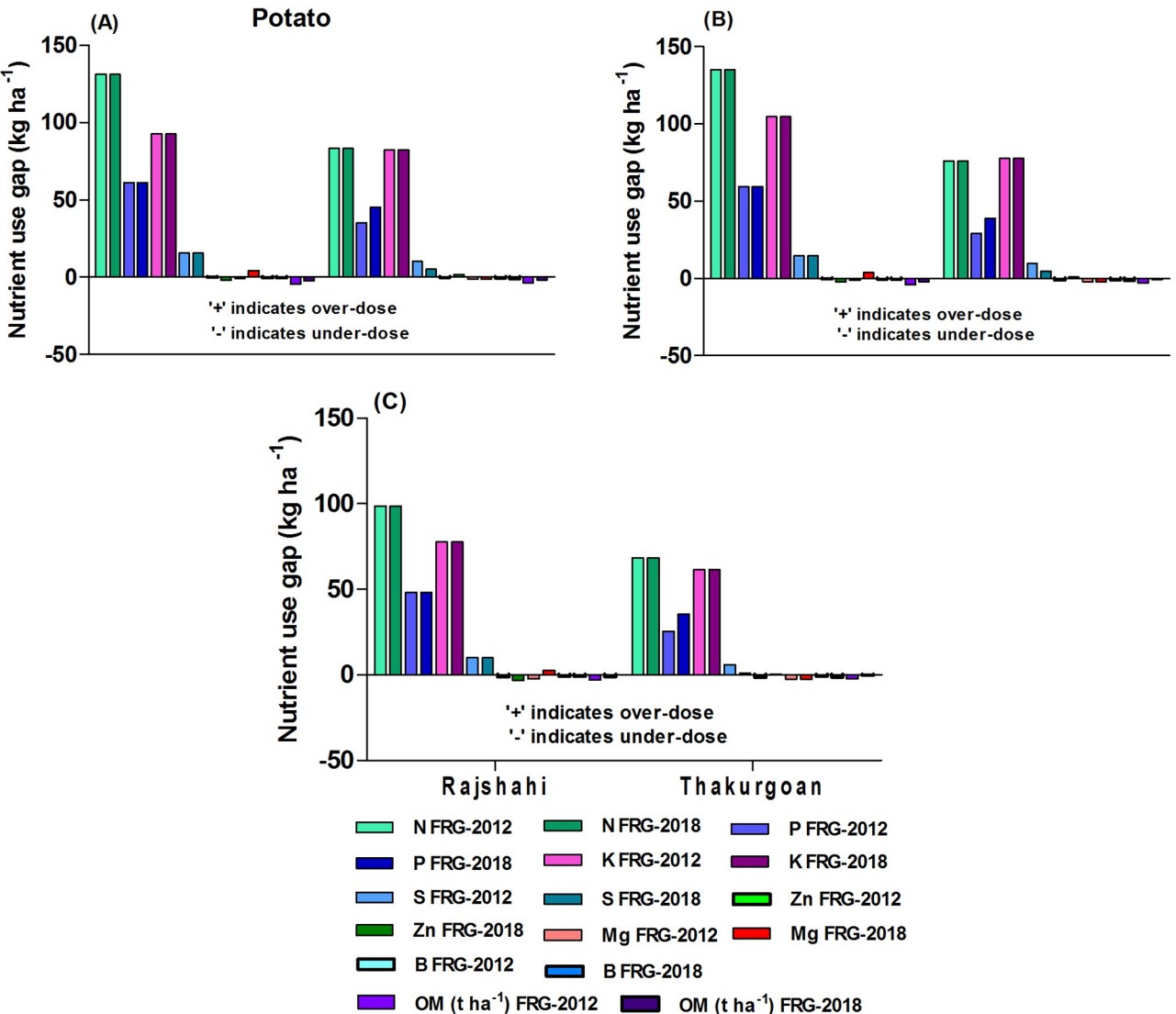

**Fig 3. Farms' current nutrient use gaps relative to FRG-2012 and FRG-2018 for potato: A) large-scale farms B) medium-scale farms C) small-scale farms for the *potato-maize-monsoon rice* cropping pattern.**

recommendations (by 4.0 and 4.0 kg ha$^{-1}$, respectively) for monsoon rice relative to FRG-2012 indicates that farm' especially large-scale farms of Rajshahi and Thakurgoan districts need to decrease rates of NPK (by 28.8, 20.5 & 17.4 and 23.9, 15.5 & 12.9 kg ha$^{-1}$, respectively) use in monsoon rice (Fig 5A). The result also indicates that all categories of farms need to increase S and Zn rates in monsoon rice relative to FRG-2018.

In the whole cropping season, we found that relative to FRG-2012 all categories of farms used around 19%, 37% and 35% over-doses of NPK, respectively in the *potato-maize-monsoon rice* cropping pattern but used under-doses of S (43%) (see S4 Table). The level of over-doses increased for P and K (86% and 44%, respectively) and the level of under-doses decreased for S (14%) according to FRG-2018 recommendations while N use gap remained same. Similar to *irrigated rice-fallow-monsoon rice* cropping pattern, the level of over-doses increased with increase of farm size.

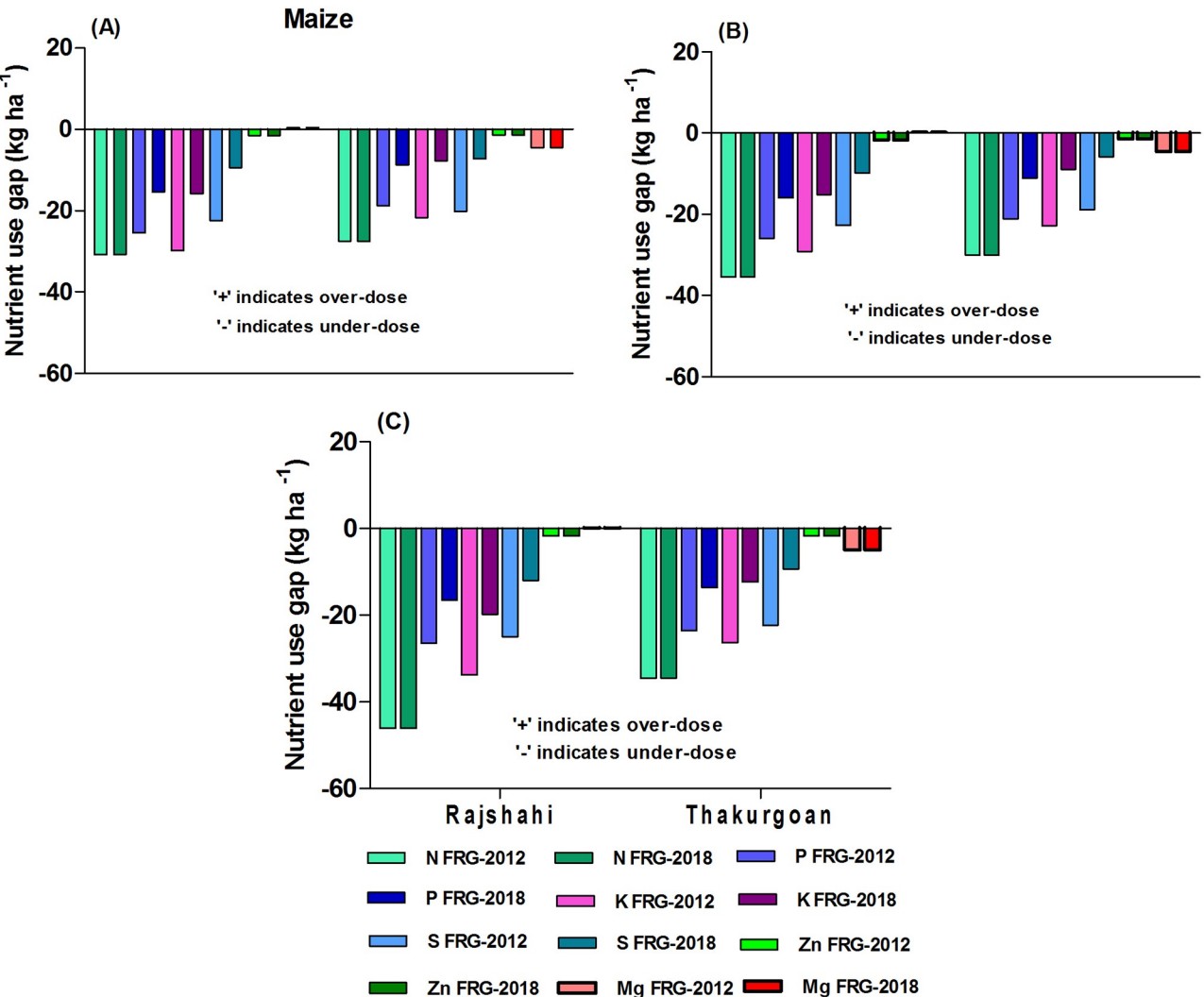

**Fig 4. Farms' current nutrient use gaps relative to FRG-2012 and FRG-2018 for maize: A) large-scale farms B) medium-scale farms C) small-scale farms under** *potato-maize-monsoon rice* **cropping pattern.**

## Financial loss for using over-dose of nutrients

Calculation of the financial loss for using over-dose of nutrients was based on FRG recommendations. Since farms used over-dosed of NPK under selected cropping patterns relative to FRG recommendations, the over-use rates of N, P and K (kg ha$^{-1}$) were converted to equivalent rates of urea, TSP and MoP (kg ha$^{-1}$), respectively to estimate the financial loss of both the farms and government as a consequence. The financial loss of the farms and the government were 25.6 and 50.4 USD ha$^{-1}$, respectively in the *irrigated rice-fallow-monsoon rice* cropping pattern relative to FRG-2012 recommendations (Table 4). In the *potato-maize-monsoon rice* cropping pattern losses were almost 3-fold higher (83.7 and 141.1 USD ha$^{-1}$, respectively) in the study areas of EGP (Table 4). Both the farms and the government are losing treasury as per the FRG-2018 recommendations under both the selected patterns. The financial loss was very high for large- and medium-scale farms as compared to small-scale farms. Similarly, large- and

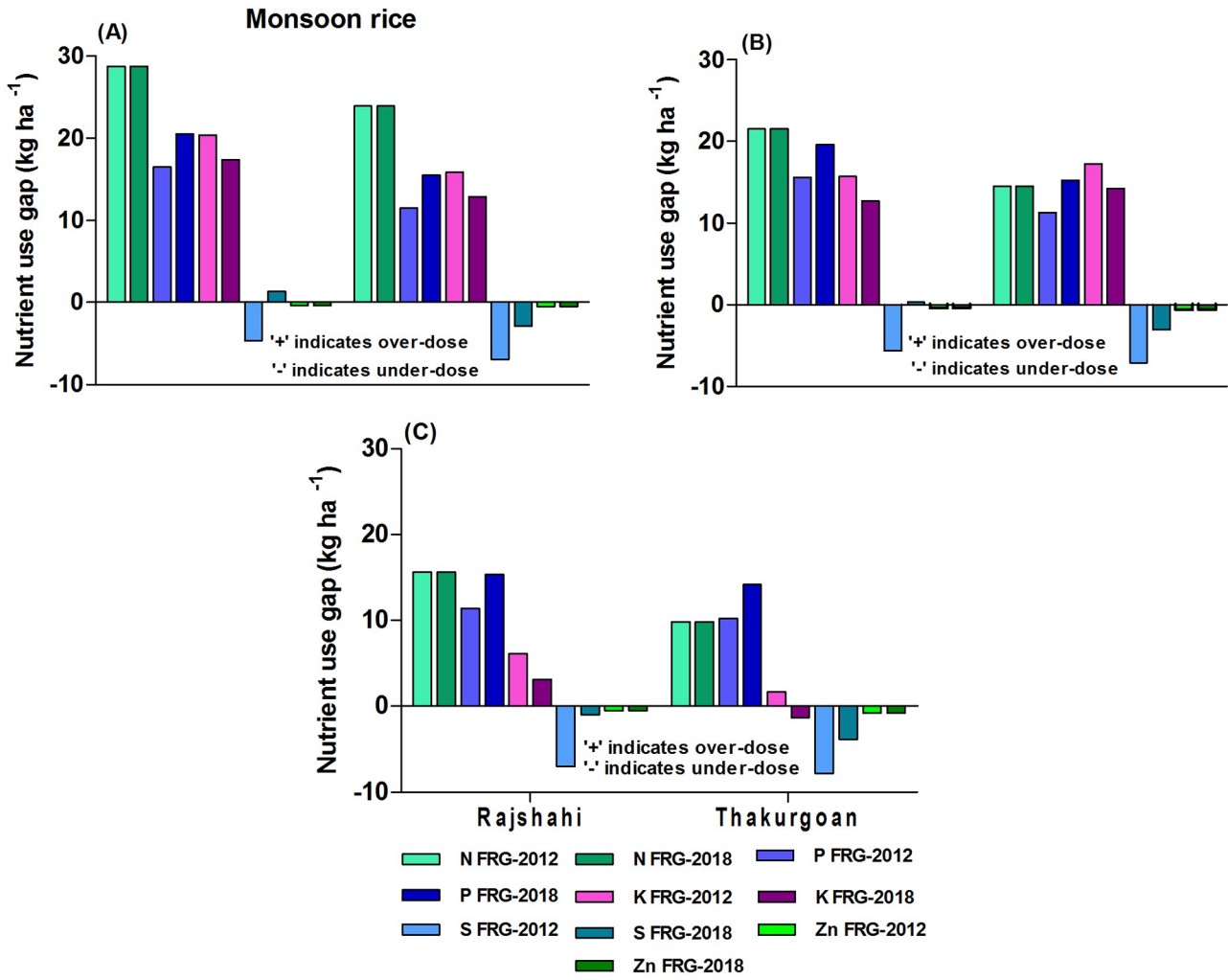

**Fig 5. Farms' current nutrient use gaps relative to FRG-2012 and FRG-2018 for monsoon rice: A) large-scale farms B) medium-scale farms C) small-scale farms in the *potato-maize-monsoon rice* cropping pattern here.**

medium-scale farms were more responsible for Government treasury losses than the small-scale farms (Table 4).

## Yield difference among different levels of nutrient users

We estimated the yield difference among different levels of nutrient users relative to FRG under selected cropping patterns. Since there was no farmer who exactly used balanced nutrient rates, ±5% deviation of total nutrient use rates from recommendations was treated as complying with the recommended dose under the *irrigated rice-fallow-monsoon rice* cropping pattern. By contrast, ±10% deviation from recommended nutrient dose was adopted for *potato-maize-monsoon rice* cropping pattern as per FRG-2018 recommendations as sample size was negligible for ±5% deviation. Indeed, for *potato-maize-monsoon rice* cropping pattern, there were no farms-applying less than the recommended dose.

The result indicates that relative to FRG-2012 recommendations, farms using the recommended dose received significantly higher irrigated rice and monsoon rice yields (0.6 and 0.6 t

**Table 4. Financial loss (USD ha$^{-1}$) of the farms and Government from overuse of NPK nutrients under rice-based cropping patterns in the study areas of the Eastern Gangetic Plain.**

| Fertilizer | Cropping patterns | | | | | | | | | | | |
|---|---|---|---|---|---|---|---|---|---|---|---|---|
| | Irrigated rice-fallow-monsoon rice | | | | | | Potato-maize-monsoon rice | | | | | |
| | Overused fertilizer rates as per FRG-2012 (kg ha$^{-1}$) | Overused fertilizer rates as per FRG-2018 (kg ha$^{-1}$) | Farmers loss as per FRG-2012 (USD ha$^{-1}$) | Farmers loss as per FRG-2018 (USD ha$^{-1}$) | Govt. loss as per FRG-2012 (USD ha$^{-1}$) | Govt. loss as per FRG-2018 (USD ha$^{-1}$) | Overused fertilizer rates as per FRG-2012 (kg ha$^{-1}$) | Overused fertilizer rates as per FRG-2018 (kg ha$^{-1}$) | Farmers loss as per FRG-2012 (USD ha$^{-1}$) | Farmers loss as per FRG-2018 (USD ha$^{-1}$) | Govt. loss as per FRG-2012 (USD ha$^{-1}$) | Govt. loss as per FRG-2018 (USD ha$^{-1}$) |
| **A. Large-scale farms** | | | | | | | | | | | | |
| Urea | 81.6 | 16.5 | 15.4 | 3.1 | 13.4 | 2.7 | 226.5 | 226.5 | 42.6 | 42.6 | 37.3 | 37.3 |
| TSP | 135.0 | 136.5 | 34.9 | 35.3 | 79.4 | 80.3 | 201.0 | 296.0 | 52.0 | 76.6 | 118.2 | 174.1 |
| MoP | 59.0 | 7.0 | 10.4 | 1.2 | 18.7 | 2.2 | 159.4 | 181.4 | 28.1 | 32.0 | 50.6 | 57.6 |
| Total loss (USD ha$^{-1}$) | | | 60.7 | 39.7 | 111.6 | 85.2 | | | 122.8 | 151.3 | 206.2 | 269.1 |
| **B. Medium-scale farms** | | | | | | | | | | | | |
| Urea | 57.3 | | 10.8 | | 9.4 | | 197.3 | 197.3 | 37.1 | 37.1 | 32.5 | 32.5 |
| TSP | 120.5 | 122.0 | 31.2 | 31.6 | 70.9 | 71.8 | 171.0 | 266.0 | 44.3 | 68.8 | 100.6 | 156.5 |
| MoP | 41.2 | | 7.3 | | 13.1 | | 161.6 | 183.6 | 28.5 | 32.4 | 51.3 | 58.3 |
| Total loss (USD ha$^{-1}$) | | | 49.2 | 31.6 | 93.4 | 71.8 | | | 109.9 | 138.4 | 184.4 | 247.3 |
| **C. Small-scale farms** | | | | | | | | | | | | |
| Urea | 5.9 | | 1.1 | | 1.0 | | 121.1 | 121.1 | 22.8 | 22.8 | 19.9 | 19.9 |
| TSP | 44.0 | 45.5 | 11.4 | 11.8 | 25.9 | 26.8 | 113.0 | 208.0 | 29.2 | 53.8 | 66.5 | 122.4 |
| MoP | -8.2 | | - | | | | 87.8 | 109.8 | 15.5 | 19.4 | 27.9 | 34.9 |
| Total loss (USD ha$^{-1}$) | | | 12.5 | 11.8 | 26.9 | 26.8 | | | 67.5 | 96.0 | 114.3 | 177.2 |
| **D. All farms** | | | | | | | | | | | | |
| Urea | 26.04 | | 4.9 | | 4.3 | | 150.6 | 150.6 | 28.3 | 28.3 | 24.8 | 24.8 |
| TSP | 72.5 | 74.0 | 18.8 | 19.2 | 42.6 | 43.5 | 136.0 | 231.0 | 35.2 | 59.8 | 80.0 | 135.9 |
| MoP | 10.8 | | 1.9 | | 3.4 | | 114.4 | 136.4 | 20.2 | 24.1 | 36.3 | 43.3 |
| Total loss (USD ha$^{-1}$) | | | 25.6 | 19.2 | 50.4 | 43.5 | | | 83.7 | 112.2 | 141.1 | 204.0 |

(Authors own calculation based on survey result)

(1 USD = 85 Taka)

*Conversion factor: urea = kg N x 2.17, TSP = kg P x 5.0 and MoP = kg K x 2.0 [29]

Maximum retail price (fixed by the Govt.) of per kg fertilizer (with subsidy) for farmers: Urea = $ 0.19, TSP = $ 0.26 and MoP = $ 0.18

Dealer purchase price (fixed by the Govt. of Bangladesh) of per kg fertilizer (with subsidy): Urea = $ 0.16, TSP = $ 0.24 and MoP = $ 0.15

Govt. procurement price of per kg fertilizer: Urea = $ 0.33, TSP = $ 0.82 and MoP = $ 0.47

Govt. subsidy of per kg Fertilizer (for imported Fertilizers): Urea = $ 0.16, TSP = $ 0.59 and MoP = $ 0.32

Source: MoA, Bangladesh, and Bangladesh Chemical Industries Corporation (BCIC), July, 2021.

ha$^{-1}$, respectively) than under-user farms, but significant lower yields (0.9 and 0.2 t ha$^{-1}$, respectively) than the over-user farms (Table 5). The nutrient use rates of the over-user category farms relative to FRG-2012 had already adjusted towards the new recommendations of FRG-2018, could be the reason of getting higher rice yields over recommended dose user

**Table 5. Yield difference among different levels of nutrient users in the cropping season relative to Government-endorsed recommendations under diverse rice-based cropping patterns.**

| Cropping pattern | Crop | As per FRG-2012 recommendations | | | | | As per FRG-2018 recommendations | | | | |
|---|---|---|---|---|---|---|---|---|---|---|---|
| | | Recommended dose user (t ha$^{-1}$) | Under-user (t ha$^{-1}$) | Over-user (t ha$^{-1}$) | Yield difference (t ha$^{-1}$) | | Recommended dose user (t ha$^{-1}$) | Under-user (t ha$^{-1}$) | Over-user (t ha$^{-1}$) | Yield difference (t ha$^{-1}$) | |
| | | (A) | (B) | (C) | (A-B) | (A-C) | (A) | (B) | (C) | (A-B) | (A-C) |
| **Irrigated rice-fallow-monsoon rice** | Irrigated rice | 5.6 (0.677) | 5.0 (0.535) | 6.5 (0.443) | 0.6*** (t = 4.288) | -0.9*** (t = -7.922) | 6.6 (0.443) | 5.2 (0.632) | 6.4 (0.422) | 1.4*** (t = 13.554) | 0.2** (t = 2.589) |
| | Monsoon rice | 4.4 (0.384) | 3.8 (0.432) | 4.6 (0.287) | 0.6*** (t = 5.761) | -0.2** (t = -2.423) | 4.7 (0.198) | 4.0 (0.487) | 4.5 (0.317) | 0.7*** (t = 9.170) | 0.2*** (t = 4.013) |
| | Sample size (n = 198) | n = 26 | n = 69 | n = 103 | | | n = 47 | n = 96 | n = 55 | | |
| **Potato-maize-monsoon rice** | Potato | 27.9 (2.098) | | 27.3 (2.979) | | 0.6 (t = 1.137) | 28.1 (2.216) | | 27.2 (2.962) | | 0.9 (t = 1.599) |
| | Maize | 5.5 (0.381) | | 5.1 (0.392) | | 0.4*** (t = 4.918) | 5.5 (0.381) | | 5.1 (0.392) | | 0.4*** (t = 4.943) |
| | Monsoon rice | 4.7 (0.375) | | 4.6 (0.390) | | 0.1 (t = 1.042) | 4.8 (0.356) | | 4.6 (0.394) | | 0.2* (t = 1.684) |
| | Sample size (n = 132) | n = 32 | | n = 100 | | | n = 36 | | n = 96 | | |

Notes: Values in the parenthesis indicates standard deviation of mean crop yield. Positive value of yield difference denotes recommended dose user received higher crop yields than over-user or under-user. The t-test was used to compare the yield difference between two categories of nutrient users.

***, ** and * indicates 1%, 5% and 10% level of significance.

Since we did not find sufficient sample size for ±5% deviation under *potato-maize-monsoon rice* cropping pattern as per FRG-2018 recommendations, ±10% deviation of nutrient user from recommended nutrient dose was treated as recommended dose user.

categories. On the other hand, the result relative to FRG-2018 indicates that recommended dose user category of farms received significant higher rice yields over both the over- and under-user categories of farms under both the selected cropping patterns. Although recommended nutrient user farms relative to both FRG-2012 and FRG-2018 received slightly higher yield in potato (0.6 and 0.9 t ha$^{-1}$, respectively) as compared to over-user categories of farms, but yield difference between two categories was not statistically significant. The result also shows that relative to both FRG-2012 and FRG-2018 recommended-dose user farms got significantly higher maize yield (0.4 and 0.4 t ha$^{-1}$, respectively) than the over-user category of farms under *potato-maize-monsoon rice* cropping pattern (Table 5).

## Farm income difference among different levels of nutrient users

The result of farm income difference among different levels of nutrient user is presented in the Table 6. The result shows that over-user category farmers relative to FRG-2012 received higher farm income compared to the recommended dose user category as this category of farmers had already adjusted to the increased nutrient recommendations of the new FRG-2018. On the other hand, the t-test confirmed that relative to FRG-2018, the recommended dose user category farmers received significant higher farm income compared to under-user and over-user categories farmers under the selected cropping patterns. Under *irrigated rice-fallow- monsoon rice* cropping pattern, the result shows that relative to FRG-2018 the recommended user category farmers received 428.9 and 105.4 USD ha$^{-1}$ higher farm income from under-user and over-user categories, respectively while it was 204.3 USD ha$^{-1}$ higher from over-user category farmers under *potato-maize-monsoon rice* cropping pattern (Table 6).

**Table 6. Farm income (USD ha$^{-1}$) difference among different levels of nutrient users in crops relative to government-endorsed recommendations under diverse rice-based cropping patterns.**

| Cropping pattern | As per FRG-2012 recommendations | | As per FRG-2018 recommendations | |
|---|---|---|---|---|
| | Farm income difference between recommended dose user and under-user | Farm income difference between recommended dose user and over-user | Farm income difference between recommended dose user and under-user | Farm income difference between recommended dose user and over-user |
| Irrigated rice-fallow- monsoon rice | 290.1*** (t = 6.9186) | -266.9*** (t = -6.3733) | 428.9*** (t = 14.0243) | 105.4*** (t = 3.0455) |
| Potato-maize-monsoon rice | | 155.6** (t = 2.0650) | | 204.3*** (t = 2.8589) |

Note:

*** , ** and * indicates 1%, 5% and 10% level of significance.

Positive value of farm income denotes recommended dose user received higher farm income over over-user or under-user. Since we did not find sufficient sample size for ±5% deviation under *potato-maize-monsoon rice* cropping pattern as per FRG-2018 recommendations, ±10% deviation of nutrient user from recommended nutrient dose was treated as recommended dose user.

## Discussion

The government-endorsed fertilizer recommendations are presumed to provide the best guidelines for farmers by which optimum crop yield and economic returns can be achieved. However, the size and factors influencing gaps between recommended nutrient use and farmers' actual use in diverse rice-based cropping patterns have rarely been examined. The present study, for the first time, documents farmers' current nutrient use gaps under diverse rice-based cropping patterns in the NWRB. The key reasons associated with farmers' current nutrient use gaps under rice-based cropping patterns are discussed in turn.

### Farm size is a major determinant of nutrient gaps

The level of over-dose was positively related with farm size. Large- and medium-scale farms are financially better off and received more credit access relative to small-scale farms (Table 5). As small-scale farms of NWRB are financially less solvent than the other parts of the country [30], during FGD we found that they are more interested to use on-farm resources such as OMs as a low cost option for nutrient supply to minimize their production cost. Similar variation of OMs use according to farm size was also observed in different parts of Nepal [12]. Farms who use high OM rates usually decrease their rates of chemical fertilizer application [31]. Although large- and medium-scale farms received more extension facilities compared to small-scale farms (Table 5), they used high over-dose of nutrients because of more ability to purchase subsidized fertilizers, while they had lower use of OMs. Farm size also acts as key determinant of adopting modern technologies like fertilizer in China and Sub-Sahara African (SSA) countries [6, 32].

Although the government of Bangladesh follows universal subsidy policy where price of NPK-containing fertilizers (per kilogram) is same for all categories of farmers [5], large- and medium-scale farms enjoy almost 1.1 and 1.6 times higher benefit of current subsidy policy compare to small-scale farms due to their greater bargaining and purchasing capability, and better awareness of fertilizer price information [33]. On the other hand, 72% of small-scale farms reported no knowledge of the magnitude of government fertilizer subsidy [26]. Similar situation also found in the SSA countries where large farms acquired more subsidized fertilizer as compared to smallholder farmers [34]. This indicates that whatever the aims of fertilizer subsidy schemes, the current design does not ensure equitable benefits among all farm sizes.

Exploring alternative designs for fertilizer subsidy schemes or alternative incentives for farms, regardless of size, to equitably access such benefits, would seem to be a useful public policy goal.

The key barriers for small-scale farms to use of optimal levels of nutrients are lack of relevant knowledge and skills, lack of sufficient working capital, and higher price of fertilizers (claimed by local fertilizer sub-dealers) during peak periods of production. Similar conclusions were found by [19, 26] for the farmers of NWRB. However, unlike the present study the previous ones considered only chemical fertilizers as a source of nutrients (excluding organic manures like decomposed cow-dung, poultry litter, crop residue etc.) and examined fertilizer gaps for single crops rather cropping patterns. In the present study, we addressed all those limitations and for the first time assessed farmers' current nutrient use gaps under diverse rice-based cropping patterns covering the full range of recommended macronutrients and micronutrients.

## Farmer investment in high yield and profit-potential crops

In the EGP, profit and yield potential play crucial roles in the choice of nutrient rates for crops under diverse cropping patterns [18]. About 99% of the irrigated rice area grows high-yielding varieties in Bangladesh and farm level yield potential is almost 40% higher than the monsoon rice [24]. Using long term experimental data [35], found that NPKS and Zn increased mean grain yield of rice by 88% over the control in the dry season in Bangladesh, but only by 37% in the wet season. The higher yield potential and yield responses may account for relatively higher over-dose of NPK in irrigated rice than the monsoon rice as per FRG-2012 in the *irrigated rice-fallow-monsoon rice* cropping pattern.

Similar to irrigated rice, yield and profit potentiality of potato is much higher relative to subsequent maize and monsoon rice in the *potato-maize-monsoon rice* cropping pattern [18]. The potato farmers of the study areas reported no training or limited training experience. As a result, peer farmers and fertilizer dealers played key roles in advising on the fertilizer rates for potato which could account for high rates of nutrients applied to potato crops relative to both FRGs. [36] reported that farmers who received advice from fertilizer dealers used 11–17% higher N rates over non-receiver. Most of the high-value potato crops are cultivated for sale in the EGP. Indeed, despite limited working capital and credit access, small-scale farms also chose to invest on fertilizers to get higher potato tuber yield, albeit at lower rates than the large- and medium-scale farms. Even in the vulnerable coastal zone of India where majority of the smallholder farmers show risk-aversion behavior, they spend almost 2 times more than the cost required for balanced nutrient use for cultivating potato in experimental plots [37]. Similar willingness to invest on fertilizer for profit potential crops by farmers is also found in other parts of the world [38, 39].

## Change in nutrient use pattern and nutrient use gaps

Prior to 2010, due to N fertilizer subsidy policy, there was less use of P and K on crops relative to N, which contributed to unbalanced nutrient rates [40]. After that, the subsidy on P and K-containing fertilizers changed the NPK ratio from 10:1.5:1.3 in 2008–09 to 10:4.6:2.8 in 2013–14 [40]. During FGD, we found that the current overdoses of P appear to be related to a continuing belief among farmers that "higher P provides higher yield". Furthermore there was almost a 120% increase in DAP fertilizer use in 2018–19 from 2011–12 in Bangladesh due to the introduction of the subsidy [17]. As DAP fertilizer contains both N and P, disproportionate use of DAP fertilizer is one of the key reasons of recent overuse of P in both rice and non-rice crops and underuse of N rates especially in rice crops relative to newly released FRG-2018.

The government of Bangladesh has announced an additional $ 94.4 million subsidy to further reduce DAP fertilizer price to 16 Tk. per kg from Tk. 25 per kg [41]. The level of P over-dose could be increased further if this subsidy scheme is not well targeted. Crops usually absorb 15–25% of the added amount of P and the rest remains in the soil where it progressively increases soil test P levels over time [15]. For that reason, cropping pattern-based research experiments often show negative balance of N, but positive balance of P in Bangladesh indicating a continuous accumulation of P in the soil [42].

All categories of farms used high rates of N for potato crops. Farmers reported in the FGD that higher doses of N than that recommended in the FRGs are needed to achieve maximum potato tuber yield. That farmers used almost double the recommended N dose for potato (this study) as they did with watermelon in a previous study [18], suggesting a lack of trust in recommendations as one of the barriers of balanced nutrient use practices in the EGP. Similar conclusions were reached from the Government of India soil health card (SHC) programme, introduced in 2015, to promote farm level balanced fertilizer use [5]. The programme failed to achieve widespread practice change for a range of reasons especially for Small-scale farms. It was concluded that farmers were unable to understand the scientific information provided, lacked confidence in the SHC fertilizer recommendations and had credit constraints that limited their use of recommended rates [13]. Even in Bangladesh, extension activities related to modern technologies, including recommended fertilizer use, often remained ineffective because of less focus on the Small-scale farms who comprise a majority of farms [43]. So, pilot scale demonstrations on recommended fertilizer doses should assist farmers replace current practices with balanced nutrients provided farmers have a close engagement with the demonstrations on farms.

In the study areas, during FGD it was apparent that lack of awareness about K fertilizer and more emphasis on urea and phosphate fertilizers were the key reasons for under-doses of K in rice crops despite low price of K-containing fertilizer in Bangladesh. Although farmers could reduce K-dose by 20–40% in following crops through incorporating 2–4 t ha$^{-1}$ crop residues in the soil [15], much of the crop residue is used as animal feed or as fuel for household cooking, or thrown outside the crop field (especially potato and maize residues) to prepare the field for the next crop. While [40, 42] reported a decreasing trend of the N-K ratio in Bangladesh, K use is far below the requirements based on soil test based K-recommendations and crop demand.

All farmers, but especially small-scale farms, used under-doses of S in rice crops but used over-dose of S in potato under the selected cropping patterns. The over-dose of S in potato indicates that cropping season, crop types and profit potentiality could be the key factors in deciding the rates of S in crops. In Bangladesh, S content of rainfall and irrigation water increased from 3.0 mg L$^{-1}$ and 1.0 mg L$^{-1}$ respectively in 1993 to 4–5 mg L$^{-1}$ and 1–2 mg L$^{-1}$ respectively in 2009 due to rapid industrialization and urbanization [35] indicating that rainfall and irrigation water are acting as an increasing source of S for crops [44]. For that reason, S deficiency symptoms may be absent, especially in monsoon rice. [18] also reported similar kinds of result for S especially in rice crops and pre-monsoon maize in NWRB.

Under-doses of Mg, Zn and B relative to the FRG-2012 and FRG-2018 were widespread. Most of the farmers in the survey appear to be unaware of the importance of using micronutrients [42]. Crops residue and OMs were the major sources of micronutrients for irrigated rice especially for small-scale farms (see S2 and S3 Tables). Although farms used relatively high rates of micronutrients in potato, it was not adequate in the absence of sufficient crop residue retention and OMs use especially for small-scale farms. On the other hand, small-scale farms reported in the FGD that micronutrients use in waterlogged condition is a wastage of money due to its lack of effectiveness. But, micronutrients especially Zn significantly increase monsoon rice grain yield [45]. As farmers used very low amounts of OMs in pre-monsoon and

monsoon crops (see S2 and S3 Tables), maize and monsoon rice were under-dosed in Mg, Zn and B supply without sufficient use of chemical fertilizers.

Like Bangladesh, large areas of India and Nepal also experience micronutrient deficits due to improper micronutrients use practices especially by the smallholder farmers [5, 46]. Some State Government of India are providing up to 50% micronutrient subsidies to boost micronutrient use by the farms [5]. Currently, there is no such policy scheme in Bangladesh to stimulate micronutrient use. As the price per unit weight of Mg, Zn, and B containing fertilizers is very high, farmers especially on small-scale farms prefer low unit-cost NPK fertilizer without considering the rate required of each [18]. Further, lack of extension and motivational activities by the Government-appointed extension agents were also identified as major cause of Mg, Zn and B underuse in the FGD.

## Farmers' perception on residual nutrients and regional factors

During FGD and KII, we noted two important perceptions of farmers using high under-dose of nutrients in monsoon rice and maize especially by the small-scale farms following irrigated rice and potato, respectively. First, farmers believed that carry over of unused fertilizers applied to irrigated rice and potato would supply required rates of nutrients for monsoon rice and maize, respectively. Secondly, excessive use of fertilizers could result in lodging of pre-monsoon maize and monsoon rice before grain maturity because of strong wind and rainfall. Nutrient adjustment between two crops in the same cropping pattern is very common practice in Bangladesh, where high yield and profit-potential crop are afforded priority compared to low yield and low profitability crops so that farmers can adjust their cost of production. [47–49] also reported similar types of nutrient use pattern and cost adjusting tendency in *potato-maize* cropping pattern in the northern Bangladesh, but did not examine the gap between farmers' current nutrient use rates and government-endorsed recommendations.

All categories of farms used higher over-doses of NPK in monsoon rice under the *potato-maize-monsoon rice* cropping pattern than that of monsoon rice cultivated in the *irrigated rice-fallow-monsoon rice* cropping pattern. Both potato and maize are nutrient demanding crops and withdraw high amounts of nutrients from the soil [48]. Moreover, low nutrient application rates in the previous maize crop might create high nutrient deficiency in the soil, which could be the reason for using high rates of NPK in monsoon rice under *potato-maize-monsoon rice* cropping pattern. However, there is still limited experimental evidence regarding the impacts of such nutrients use practices on monsoon rice yield for crop profit in the *potato-maize-monsoon rice* cropping system.

By contrast with P, K, S, Mg, B and Zn which have high residual effect on subsequent crops, N has low residual effects on the subsequent crops as 60% of applied N can be lost through leaching, denitrification and volatilization [15]. Recent studies suggest that 15–20% of urea N was lost from irrigated rice in northern Bangladesh as ammonia [4]. For the nutrient with residual effects, farmers could reduce rates by 30–50% in the following crops depending on crop types, duration and season of the crop cultivation [15]. Indeed, the FRG allows for residual value of nutrients in the pattern-based nutrient recommendations. Hence, the farmers' current nutrient use practices involving high over-dose of nutrients in a high value crop followed by an under dose in the following crop may be justifiable for P, K, S, Mg and Zn but not for N which has no or very low residual effects. This is an issue where training for farmers could reduce costs of production and increase profits.

The result showed that farms of Rajshahi district used high over-dose of nutrients relative to Mymensingh and Thakurgoan districts in both the selected patterns. Variation in physiography, eco-system and environment among regions might be the reasons for using nutrients

differently. Rajshahi is situated in the High Barind Tract region and has lower average annual rainfall (1581 mm) than the Mymensingh (2500 mm) and Thakurgoan (2218 mm) districts [50]. Crops grown in Rajshahi district need frequent irrigation water on the predominantly highland and medium highland fields. Moreover, low soil nutrient status (N, P, S, and B) of Rajshahi district [15] justifies higher rates of nutrients relative to other parts of the country [18].

## Yield, farm income and financial losses scenario

It is clear from the overuse of NPK especially in potato crop that farmers' current nutrient use mainly focused on maximizing crop yields rather than optimizing economic use of nutrients. Indeed, average yield of irrigated rice and potato farmers in NWRB are 4.1 and 18.2 t ha$^{-1}$, respectively [21] which are far below the attainable yields (7.5 ± 0.75 and 30 ± 3.0 t ha$^{-1}$, respectively) suggested in the FRG. Similarly, the yield gap of maize farmers (1.8 t ha$^{-1}$) under *potato-maize-monsoon rice* cropping pattern is still very high compared to experimental plot in the northern Bangladesh [48]. High over-dose of NPK used in potato did not significantly increase potato tuber yield, and reduces farming profitability and increases environmental degradation [51]. Our study indicated that farmers using close to recommended dose relative to newly released FRG-2018 received higher irrigated rice, potato and maize yields over the farmers who used over or under doses of nutrients. Similarly, reduction of excess N use by 50% could increase household income by 4–15% in China [52]. Hence, there is a substantial scope to reduce current crop yield gaps and promote farm income through balanced nutrient use in the EGP.

The blanket fertilizer subsidy to all categories of farms in the EGP has increased fertilizer use intensity especially the use of P and K fertilizers, and increased crop productivity and farming efficiency [33] but requires substantial investment by the governments. For example, rice production has increased almost 46% in Bangladesh from 2000 to 2020 [16] partly due to increased use of fertilizer along with expansion of irrigation provision and adoption of modern cultivars. Recently, the government of Bangladesh spent 823.5 million US$ in subsidies for fertilizer [53] but a threefold increase is expected in 2022 due to escalating world fertilizer prices. On the other hand, food price could increase by 2.2 times in Bangladesh [54] if the government removed the current blanket fertilizer subsidy which ultimately could reduce the food accessibility of millions of poor people. Our findings suggest that an education programme for farmers to use recommended fertilizer rates could boost national grain production for food security and produce greater returns to farmers and the government from the fertilizer subsidy. Considering the progressive decline in area of cultivable land, the ongoing growth of population and the necessity to provide for their food security and the specific livelihood needs of rural marginal poor people, as well as soil health and environmental quality, there is an urgent need for using balanced nutrients in the EGP.

## Conclusion

Up to 80% of farmers of the study areas are currently using highly unbalanced nutrients for crops relative to FRG recommendations. Unbalanced fertilizer use is eroding farm profits, adding to the national Government's fertilizer subsidy costs and is a drag on improving food security. Extension agents need to be more focused especially on the small-scale farms to increase the rates of N, K, S, Mg, Zn, B use and also on medium- and large-scale farms to reduce the excess rates of P use. Extension and motivational work need to be strengthened for non-rice crops like potato to reduce the unprofitable use of highly subsidized NPK nutrients, lower production costs, and to increase crop yields. Programmes to ensure adequate supply of

OM in the local markets as well as motivation towards its use especially on large-scale farms could decrease dependency on chemical fertilizers. The newly published FRG needs to be promoted through extension programmes and follow up visits so that farmers gain confidence to use FRG-based recommended nutrients to reduce yield gaps and environmental losses.

## Supporting information

**S1 Table. Nutrient composition of different nutrient sources.**
(DOCX)

**S2 Table. Rates of nutrient inputs (kg ha$^{-1}$) used in the *irrigated rice-fallow-monsoon rice* cropping pattern.**
(DOCX)

**S3 Table. Rates of nutrient inputs (kg ha$^{-1}$) used in the *potato-maize-monsoon rice* cropping pattern.**
(DOCX)

**S4 Table. Farmers current NPKS use gap in cropping season under different rice-based cropping patterns in the northern Bangladesh.**
(DOCX)

**S5 Table. Current nutrient use rates (kg ha$^{-1}$) of different categories of farmers under *irrigated rice-fallow-monsoon rice* cropping pattern.**
(DOCX)

**S6 Table. Current nutrient use rates (kg ha$^{-1}$) of different categories of farmers under *potato-maize-monsoon rice* cropping pattern in the selected study areas.**
(DOCX)

**S7 Table. Recommended nutrient rates (kg ha$^{-1}$) of irrigated rice, monsoon rice, potato and maize grown under *irrigated rice-fallow-monsoon rice* cropping pattern and *potato-maize-monsoon rice* cropping patterns.**
(DOCX)

**S1 Dataset.**
(XLSX)

## Acknowledgments

The authors gratefully acknowledge the "Nutrient Management for Diversified Cropping in Bangladesh (NUMAN)" project personals and Bangladesh Agricultural Research Council (BARC) for all kinds of technical supports.

## Author Contributions

**Conceptualization:** Md. Shofiqul Islam, Mohammad Jahangir Alam.

**Formal analysis:** Md. Shofiqul Islam.

**Funding acquisition:** Richard W. Bell.

**Methodology:** Richard W. Bell, M. A. Monayem Miah, Mohammad Jahangir Alam.

**Project administration:** M. A. Monayem Miah.

**Supervision:** Richard W. Bell, Mohammad Jahangir Alam.

**Validation:** Mohammad Jahangir Alam.

**Writing – original draft:** Md. Shofiqul Islam.

**Writing – review & editing:** Richard W. Bell, M. A. Monayem Miah, Mohammad Jahangir Alam.

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
