## [Decision Letter · Decision Letter 0]

24 Feb 2022

PONE-D-22-01407Unbalanced Fertilizer Use in the Eastern Gangetic Plain: the Influence of Government Recommendations, Fertilizer Type, Farm Size and Cropping PatternsPLOS ONE

Dear Dr. Alam,

Thank you for submitting your manuscript to PLOS ONE. After careful consideration, we feel that it has merit but does not fully meet PLOS ONE’s publication criteria as it currently stands. Therefore, we invite you to submit a revised version of the manuscript that addresses the points raised during the review process.

We look forward to receiving your revised manuscript.

Kind regards,

Dibyendu Chatterjee, Ph.D.

Academic Editor

PLOS ONE

Journal Requirements:

3. Please provide additional details regarding participant consent. In the Methods section, please ensure that you have specified (1) whether consent was informed and (2) what type you obtained (for instance, written or verbal). If your study included minors, state whether you obtained consent from parents or guardians. If the need for consent was waived by the ethics committee, please include this information.

[The authors gratefully acknowledge the “Nutrient Management for Diversified Cropping in Bangladesh (NUMAN)” project and Bangladesh Agricultural Research Council (BARC) for financial and technical support.]

 [Australian Centre for International Agricultural Research (ACIAR) and Krishi Gobeshona Foundation (KGF), Bangladesh. ACIAR grant number: LWR/2016/136 & KGF grant number: (CN/FRPP):-ICP-II.]

Additional Editor Comments:

This research paper describes the reason behind the yield gap in Bangladesh. Please address the following issues: (i) to improve the synchronization between the paragraphs in Introduction, (ii) ambiguity of cropping system is to be addressed, (iii) Results need to be more specific and irrelevant information may be omitted in Results and Discussion, (iv) The authors should highlight the reason behind the higher P application (overdose) by small and marginal farmers. Also, the reason behind 16-90% over dose of NPK. (v) Lack of more detailed information about the approaches by the Government.

Reviewers' comments:

Reviewer's Responses to Questions

**Comments to the Author**

1. Is the manuscript technically sound, and do the data support the conclusions?

Reviewer #1: Yes

Reviewer #2: Partly

Reviewer #3: Yes

2. Has the statistical analysis been performed appropriately and rigorously? 

Reviewer #1: Yes

Reviewer #2: Yes

Reviewer #3: Yes

3. Have the authors made all data underlying the findings in their manuscript fully available?

Reviewer #1: Yes

Reviewer #2: Yes

Reviewer #3: Yes

4. Is the manuscript presented in an intelligible fashion and written in standard English?

Reviewer #1: Yes

Reviewer #2: Yes

Reviewer #3: Yes

5. Review Comments to the Author

Reviewer #1: The manuscript "Unbalanced Fertilizer Use in the Eastern Gangetic Plain: the Influence of Government Recommendations, Fertilizer Type, Farm Size and Cropping Patterns" is written well. It is interesting and useful for researchers, farmers and policy makers of Bangladesh.

I recommend to improve the figures quality.

Reviewer #2: Dear Author

The paper is important from the view point of policy guidelines. Farmers generally apply fertilizer based on crop requirement, market price, subsidy, climate etc. The fact that FRG 18 has higher values for NPK than 12 indicate that there were requirement for higher NPK doses for crops. Further two FRG guidelines created confusion, you must focus on only one. If we considerer farmers wise, they will apply fertilizer until they will get some good response, else not. However, they may apply more of one and less of another based on availability, crop response and amount to be invested. If there is subsidy on fertilizer, why farmers applied less of micronutrients and K. I am surprised to see that even small and marginal farmers applied over dose of P fertilizer. Explain what could be the reason? Also you told that there were in general 16-90% over dose of NPK used by different categories of farmers? What's their proportionate yield and income enhancement? Whether subsidy on fertilizer is blanket ? Or its based on certain principles say for a farmers cultivating X ha land will he get Y amount subsidy? or its subsidy on all amount they want to apply? You mentioned in line no 474 that there is lack of trust on Govt recommendation? What could be the reason? How its addressed by the concerned Dept? Whats the novelty of the study and also scientific basis? I find its more of a technology Gap or lack of awareness among farmers due to inappropriate extension activities. There should be pilot scale demonstrations on recommended doses and the farmers should be made aware by their visit and hands on in such study areas. A detail and critical analysis on impact of over or less use is required on productivity, income, envt pollution and Govt schemes on subsidy for its future rationalization. Whether Govt recommendation is based on STCR (soil test crop response), or any other scientific approaches?

THANK YOU

Reviewer #3: In this article, Islam et al. has highlighted the unbalanced fertilizer use scenario in Bangladesh with respect to govt. policies, farm size, and cropping pattern. The article is well designed, executed, discussed and concluded. It fairly widens the knowledge on causes behind yield gaps in major cropping patterns in Bangladesh. However, it requires a few minor rectifications before it can be considered for publication. Some of my main criticisms are as follows:

1. Abstract is satisfactory

2. Introduction

The introduction provides sufficient background information. However, it lacks synchrony between the paragraphs. Please improve readability and linkages between the paragraphs in this section.

There is no clear indication of objectives; instead the authors merely highlight what they were intended to.

Please check the sequence of citations……..it starts with [8]!.

3. M&M

I find ambiguity in the represented sequence of crops in the cropping patterns. A cropping pattern should represent the crops cultivated from beginning of a calendar year not from the end of it. For example, monsoon rice is always cultivated ahead of irrigated rice in a calendar year; so the cropping pattern should represent as: monsoon rice-irrigated rice-fallow. The other cropping pattern should be: maize (summer)-monsoon rice (rainy)-potato (winter). I suggest to comply with it in whole MS.

L94-102: Please provide grid information for the experimental sites.

L100: Add citation to the sampling procedure.

4. Results

Descriptive statistics: Sentences are very clumsy with poor readability. Please recast sentences to improve clarity.

In a few instances, results are discussed (e.g., L340-342), which should be avoided. Otherwise, the results are too descriptive in nature and efforts are needed to shorten its length as far as possible.

5. Discussion

L371-393: Unnecessary paragraphs and should be omitted. A part may go to M&M section.

Why have the authors placed Table 5 in discussion section? It should go to M&M section.

L433-436: The authors must not confuse between HYVs and hybrids! Be precise.

L576-591: This part of discussion seems unnecessary. These are fallout of the imbalanced fertilizer uses and a part of it is already highlighted in introduction. I would suggest to erase this part, along with environmental loss scenario in heading.

6. Conclusion is satisfactory

6. PLOS authors have the option to publish the peer review history of their article (what does this mean?). If published, this will include your full peer review and any attached files.

Reviewer #1: No

Reviewer #2: No

Reviewer #3: No

---

## [Author Response · Author response to Decision Letter 0]

18 May 2022

Response to Reviewers` Comments

Journal Name: PLOS ONE 

Manuscript Number: PONE-D-22-01407

Manuscript Title: Unbalanced fertilizer use in the Eastern Gangetic Plain: the influence of Government recommendations, fertilizer type, farm size and cropping patterns

Academic Editor

Comment: This research paper describes the reason behind the yield gap in Bangladesh. Please address the following issues: 

(i) to improve the synchronization between the paragraphs in Introduction, 

(ii) ambiguity of cropping system is to be addressed, 

(iii) (Results need to be more specific and irrelevant information may be omitted in Results and Discussion, 

(iv) The authors should highlight the reason behind the higher P application (overdose) by small and marginal farmers. Also, the reason behind 16-90% over dose of NPK. Lack of more detailed information about the approaches by the Government

Response: (i) We have added some sentences in the introduction section (line 57-58, line 74-79 and line 85-88) which improve the readability and linkages between the paragraphs in this section.

(ii) Cropping year convention used in Bangladesh consists of three major seasons - Robi season (mid-November to mid-March), Kharif-1 season (mid-March to mid-July) and Kharif-2 (mid-July to mid-November) season. The FRG sequence of crops in the cropping pattern always list the Robi season crop as first crop, so we have followed that convention. Please see the following reference which published by PLOS ONE journal: Al Mamun M.A, Nihad SAI, Sarkar M.AR, Aziz M.A, Qayum M.A, Ahmed R, et al. (2021) Growth and trend analysis of area, production and yield of rice: A scenario of rice security in Bangladesh. PLoS ONE 16(12): e0261128. doi.org/10.1371/journal.pone.0261128

(iii) We have re-written the results section as per reviewer suggestions and omitted irrelevant information from the manuscript. 

(iv) The reason behind the higher P application (overdose) by different farm sizes including small-scale farms and 16-90% over dose of NPK is described thoroughly in the sub-section 4.3. 

(v) Lack of more detailed information about the approaches by the Government is presented in the sub-section 4.3, line 476-480.

Reviewer 1: 

Comment: The manuscript "Unbalanced Fertilizer Use in the Eastern Gangetic Plain: the Influence of Government Recommendations, Fertilizer Type, Farm Size and Cropping Patterns" is written well. It is interesting and useful for researchers, farmers and policy makers of Bangladesh. I recommend to improve the figures quality

Response: We have changed all the previous figures of the manuscripts

Reviewer 2: 

Comment: Farmers generally apply fertilizer based on crop requirement, market price, subsidy, climate etc. The fact that FRG 18 has higher values for NPK than 12 indicate that there were requirements for higher NPK doses for crops. Further two FRG guidelines created confusion, you must focus on only one.

Response: The variation in nutrient gaps among the two FRGs was an important factor in this study. We have justified (sub-section 2.5) why we assessed gaps relative to both FRGs. The FRG-2012 recommendation was available for the farmers at the time of the survey. But the FRG-2012 was replaced by a new version of FRG (FRG-2018) which was available for the farmers soon after the field survey was completed. To assess the existing scenario of nutrient use gaps and to identify future nutrient use gaps we assessed both FRGs.

Comment: If we consider farmers wise, they will apply fertilizer until they will get some good response, else not. However, they may apply more of one and less of another based on availability, crop response and amount to be invested. If there is subsidy on fertilizer, why farmers applied less of micronutrients and K. I am surprised to see that even small and marginal farmers applied over dose of P fertilizer. Explain what could be the reason?

Response: This is another very important question. We have explained the reason of over-dosing and under-dosing of nutrients (including micronutrients) in the sub-section 4.3. Generally, both farmers’ perception and fertilizer subsidy are key matters in deciding fertilizer rates. Farmers perception “higher P provides higher yield” along with recent increase subsidy on P fertilizers is highly responsible for P overdosing even by the small and marginal farmers. On the other hand, lack of awareness was main barrier to optimal K use especially in rice crops. During focus group discussion (FGD), farmers reported that they are unable to use micronutrients due to higher price. Currently Govt. is providing subsidy only for NPK containing fertilizers.

Comment: You told that there were in general 16-90% over dose of NPK used by different categories of farmers? What's their proportionate yield and income enhancement?

Response: In the present study, we defined three categories of farmers: over-user, under-user and recommended dose user and then estimated the crop yield and farm income difference among different levels of nutrient user (Table 5 and Table 6). Due to small sample size especially for large and medium farmers, we did not estimate yield and income enhancement due to recommended nutrient dose use by farm sizes with nutrient use gaps (over-user and under-user). Since this was a survey-based study, the data collected was not suitable to estimate proportionate yield and income enhancement for individual nutrient over-dosing and under-dosing. But, our findings clearly indicated that Govt. endorsed recommended dose user received higher yield and income over unbalanced nutrient user (over-doing or under-dosing). 

Comment: Whether subsidy on fertilizer is blanket? Or it’s based on certain principles say for a farmers cultivating X ha land will he get Y amount subsidy? Or its subsidy on all amount they want to apply?

Response: In Bangladesh, Govt. follows universal subsidy policy which means price of NPK-containing fertilizers (per kilogram) is same for all categories of farmers what amount they wish to purchase (mentioned in the sub-section 4.1, line 407-412).

Comment: You mentioned in line no 474 that there is lack of trust on Govt. recommendation? What could be the reason? How it’s addressed by the concerned Dept.? What’s the novelty of the study and also scientific basis? I find it’s more of a technology Gap or lack of awareness among farmers due to inappropriate extension activities. There should be pilot scale demonstrations on recommended doses and the farmers should be made aware by their visit and hands on in such study areas.

Response: During FGD, farmers reported that they think Government nutrient recommendation is too conservative and they can enhance crop yield by increasing the rates of nutrients. The Governments various extension activities related to modern technologies adoption including recommended fertilizer use often remain ineffective because of less focus on the large numbers of the small and marginal farms that make the majority of farmers. In a related pilot scale project, we are engaging with farmers on recommended fertilizer doses to replace current practices with balanced nutrients by building their confidence. The existing weaknesses in the extension system will be made more apparent by the present results in the study areas. We have discussed these issues in the sub-section 4.3. 

The research gaps are now presented in the introduction section and the novelty of study is presented in the discussion section (last paragraph of sub-section 4.1, line 422-427).

Comment: A detail and critical analysis on impact of over or less use is required on productivity, income, environmental pollution and Govt. schemes on subsidy for its future rationalization.

Response: A detailed and critical analysis on impact of over or less use is presented in the sub-section 4.5. We have re-arranged the sub-section based on the comments of Reviewer-3. 

Comment: Whether Govt. recommendation is based on STCR (soil test crop response), or any other scientific approaches?

Response: Govt. recommendation is based on STCR (soil test crop response).

Reviewer 3

Comment: The introduction provides sufficient background information. However, it lacks synchrony between the paragraphs. Please improve readability and linkages between the paragraphs in this section.

Response: We have added some sentences in the introduction section that improve readability and linkages between the paragraphs in this section.

Comment: There is no clear indication of objectives; instead, the authors merely highlight what they were intended to. Please check the sequence of citations……..it starts with [8]!

Response: The aim of the study (including hypothesis) is now stated in the last paragraph of the introduction section. We have re-arranged the sequence of citations.

Comment: M&M I find ambiguity in the represented sequence of crops in the cropping patterns. A cropping pattern should represent the crops cultivated from beginning of a calendar year not from the end of it. For example, monsoon rice is always cultivated ahead of irrigated rice in a calendar year; so, the cropping pattern should represent as: monsoon rice-irrigated rice-fallow. The other cropping pattern should be: maize (summer)-monsoon rice (rainy)-potato (winter). I suggest to comply with it in whole MS.

Response: Cropping year convention used in Bangladesh consists of three major seasons - Robi season (mid-November to mid-March), Kharif-1 season (mid-March to mid-July) and Kharif-2 (mid-July to mid-November) season. The FRG sequence of crops in the cropping pattern always list the Robi season crop as first crop, so we have followed that convention.

Recent articles published by PLOS ONE followed the same convention for sequence of crops under a cropping pattern (e.g., irrigated rice (called as Boro rice)-fallow-monsoon rice (called as T. Aman rice).

Al Mamun M.A, Nihad SAI, Sarkar M.AR, Aziz M.A, Qayum M.A, Ahmed R, et al. (2021) Growth and trend analysis of area, production and yield of rice: A scenario of rice security in Bangladesh. PLoS ONE 16(12): e0261128. doi.org/10.1371/journal.pone.0261128

Comment: L94-102: Please provide grid information for the experimental sites.

Response: We have provided grid information for the study sites in the first para of sub-section 2.1.

Comment: L100: Add citation to the sampling procedure.

Response: We have added citation to the sampling procedure.

Comment: Descriptive statistics: Sentences are very clumsy with poor readability. Please recast sentences to improve clarity. In a few instances, results are discussed (e.g., L340-342), which should be avoided.

Response: We have re-written the sentences to improve clarity.

Comment: L371-393: Unnecessary paragraphs and should be omitted. A part may go to M&M section. Why have the authors placed Table 5 in discussion section? It should go to M&M section.

Response: Most of the unnecessary paragraphs (L371-393) are omitted from the discussion section. We now placed the Table 5 in the result section (Sub-section 3.1) as it is a finding of our present study.

Comment: L433-436: The authors must not confuse between HYVs and hybrids! Be precise.

Response: We have rearranged the sentences.

Comment: L576-591: This part of discussion seems unnecessary. These are fallout of the imbalanced fertilizer uses and a part of it is already highlighted in introduction. I would suggest to erase this part, along with environmental loss scenario in heading.

Response: As per your suggestion, we have discarded the paragraph (L576-591) and re-written the headings.

---

## [Decision Letter · Decision Letter 1]

15 Jun 2022

PONE-D-22-01407R1Unbalanced fertilizer use in the Eastern Gangetic Plain: The influence of Government recommendations, fertilizer type, farm size and cropping patternsPLOS ONE

Dear Dr. Alam,

Thank you for submitting your manuscript to PLOS ONE. After careful consideration, we feel that it has merit but does not fully meet PLOS ONE’s publication criteria as it currently stands. Therefore, we invite you to submit a revised version of the manuscript that addresses the points raised during the review process. Please submit your revised manuscript by Jul 30 2022 11:59PM. If you will need more time than this to complete your revisions, please reply to this message or contact the journal office at plosone@plos.org. Please include the following items when submitting your revised manuscript:A rebuttal letter that responds to each point raised by the academic editor and reviewer(s). You should upload this letter as a separate file labeled 'Response to Reviewers'.A marked-up copy of your manuscript that highlights changes made to the original version. You should upload this as a separate file labeled 'Revised Manuscript with Track Changes'.An unmarked version of your revised paper without tracked changes. You should upload this as a separate file labeled 'Manuscript'.If applicable, we recommend that you deposit your laboratory protocols in protocols.io to enhance the reproducibility of your results. Protocols.io assigns your protocol its own identifier (DOI) so that it can be cited independently in the future. For instructions see: https://journals.plos.org/plosone/s/submission-guidelines#loc-laboratory-protocols. Additionally, PLOS ONE offers an option for publishing peer-reviewed Lab Protocol articles, which describe protocols hosted on protocols.io. Read more information on sharing protocols at https://plos.org/protocols?utm_medium=editorial-email&utm_source=authorletters&utm_campaign=protocols.

We look forward to receiving your revised manuscript.

Kind regards,

Dibyendu Chatterjee, Ph.D.

Academic Editor

PLOS ONE

Journal Requirements:

Additional Editor Comments:

You have addressed all the comments and suggestions of reviewer 1 and 2. Please address some minor comments by reviewer 2.

Reviewers' comments:

Reviewer's Responses to Questions

**Comments to the Author**

1. If the authors have adequately addressed your comments raised in a previous round of review and you feel that this manuscript is now acceptable for publication, you may indicate that here to bypass the “Comments to the Author” section, enter your conflict of interest statement in the “Confidential to Editor” section, and submit your "Accept" recommendation.

Reviewer #2: All comments have been addressed

Reviewer #3: All comments have been addressed

2. Is the manuscript technically sound, and do the data support the conclusions?

Reviewer #2: Yes

Reviewer #3: Yes

3. Has the statistical analysis been performed appropriately and rigorously? 

Reviewer #2: Yes

Reviewer #3: Yes

4. Have the authors made all data underlying the findings in their manuscript fully available?

Reviewer #2: Yes

Reviewer #3: No

5. Is the manuscript presented in an intelligible fashion and written in standard English?

Reviewer #2: Yes

Reviewer #3: Yes

6. Review Comments to the Author

Reviewer #2: Dear Author

Thank you very much for addressing most of the issues. However, you must mention some assessment about how much is yield or income enhancement due to overuse of fertilizer over the recommended one by the govt.? Also give your own assessment the impact of blanket subsidy to all group of farmers and also on possible enhancement in productivity, food security and economical impact.

Thank you

Reviewer #3: The authors have successfully addressed my queries to a large extent. Now, the MS can be accepted for publication.

7. PLOS authors have the option to publish the peer review history of their article (what does this mean?). If published, this will include your full peer review and any attached files.

Reviewer #2: No

Reviewer #3: **Yes: **Sovan Debnath

---

## [Author Response · Author response to Decision Letter 1]

30 Jun 2022

Response to Reviewers` Comments

30 June 2022

Journal Name: PLOS ONE 

Manuscript Number: PONE-D-22-01407R1

Manuscript Title: Unbalanced Fertilizer Use in the Eastern Gangetic Plain: the Influence of Government Recommendations, Fertilizer Type, Farm Size and Cropping Patterns

Academic Editor 

Comment: Please review your reference list to ensure that it is complete and correct. If you have cited papers that have been retracted, please include the rationale for doing so in the manuscript text, or remove these references and replace them with relevant current references. Any changes to the reference list should be mentioned in the rebuttal letter that accompanies your revised manuscript. If you need to cite a retracted article, indicate the article’s retracted status in the References list and also include a citation and full reference for the retraction notice.

Response: We have re-checked the reference list and found all are in the reference list which we have cited in the manuscript. We have included some references (e.g. reference 54) as well as replaced some references (e.g. reference 10) in order to emphasise regional (Eastern Gangetic Plain) context as well as address the reviewers’ comments and suggestions. 

Reviewer 2

Comment 1: You must mention some assessment about how much is yield or income enhancement due to overuse of fertilizer over the recommended one by the govt.?

Response: Yield and income enhancement of different levels of nutrient user, due to balanced fertilizer use, are shown in Table 5 and Table 6, respectively which were testing for significance by the t-test. Recommended nutrient user received significant higher yield and income over under-user and over-user category. Only potato over-user famers got slightly higher yield over recommended user but it was statistical non-significant. We discussed these topics in the discussion section (line 559-571).

Comment 2: Also give your own assessment the impact of blanket subsidy to all group of farmers and also on possible enhancement in productivity, food security and economical impact.

Response: Our own assessment of the impact of blanket subsidy (both Govt. and farmers perspective) to all group of farmers is mentioned in the discussion section (line 572-586). The blanket fertilizer subsidy has increased crop productivity and farming efficiency but requires substantial investment by the Governments. Increased productivity of rice crop has positive impact on achieving food security. If Govt. removed the blanket fertilizer subsidy, the food price could rise dramatically. That’s why our suggestion was to implement balanced fertilizer use practices by the farmers that could save Govt. treasury as well as ensure food security by increasing crop productivity and farming profitability.

---

## [Editor Report · Decision Letter 2]

14 Jul 2022

Unbalanced fertilizer use in the Eastern Gangetic Plain: The influence of Government recommendations, fertilizer type, farm size and cropping patterns

PONE-D-22-01407R2

Dear Dr. Alam,

We’re pleased to inform you that your manuscript has been judged scientifically suitable for publication and will be formally accepted for publication once it meets all outstanding technical requirements.

Kind regards,

Dibyendu Chatterjee, Ph.D.

Academic Editor

PLOS ONE
---

## [Editor Report · Acceptance letter]

18 Jul 2022

PONE-D-22-01407R2 

Unbalanced fertilizer use in the Eastern Gangetic Plain: The influence of Government recommendations, fertilizer type, farm size and cropping patterns 

Dear Dr. Alam:

I'm pleased to inform you that your manuscript has been deemed suitable for publication in PLOS ONE. Congratulations! Your manuscript is now with our production department. 

Kind regards, 

on behalf of

Dr. Dibyendu Chatterjee 

Academic Editor

PLOS ONE